# nextPYP: a comprehensive and scalable platform for characterizing protein variability in situ using single-particle cryo-electron tomography

Hsuan-Fu Liu [1,5], Ye Zhou [2,5], Qinwen Huang [2], Jonathan Piland [3], Weisheng Jin[2], Justin Mandel[2], Xiaochen Du [4], Jeffrey Martin[2] & Alberto Bartesaghi [1,2,3]✉

Single-particle cryo-electron tomography is an emerging technique capable of determining the structure of proteins imaged within the native context of cells at molecular resolution. While high-throughput techniques for sample preparation and tilt-series acquisition are beginning to provide sufficient data to allow structural studies of proteins at physiological concentrations, the complex data analysis pipeline and the demanding storage and computational requirements pose major barriers for the development and broader adoption of this technology. Here, we present a scalable, end-to-end framework for single-particle cryo-electron tomography data analysis from on-the-fly pre-processing of tilt series to high-resolution refinement and classification, which allows efficient analysis and visualization of datasets with hundreds of tilt series and hundreds of thousands of particles. We validate our approach using in vitro and cellular datasets, demonstrating its effectiveness at achieving high-resolution and revealing conformational heterogeneity in situ. The framework is made available through an intuitive and easy-to-use computer application, nextPYP (http://nextpyp.app).

Cryo-electron tomography is an emerging technique used to determine the structure of proteins in their native, frozen hydrated state. By taking a series of tilted images of vitrified specimens in an electron microscope, it is possible to visualize unique pleomorphic events and macromolecular organizations that would otherwise be very difficult to reconstitute in vitro. Repetitive or frequently occurring molecules within these pleomorphic objects can be averaged in three dimensions to create higher resolution maps using single-particle cryo-electron tomography (SP-CET). This technology has been successfully applied to study biological systems reconstituted in vitro[1–4] or subcellular complexes present within intact bacteria or eukaryotic cells[5–8].

Recent advances in sample preparation using cryo-focused ion beam (cryo-FIB)[9–12] instruments combined with strategies for parallel data acquisition using beam-image shift[13–15] have increased the throughput of data collection allowing to routinely acquire datasets with hundreds of tilt series. Moreover, the introduction of platforms for sample screening to automatically navigate cryo-electron microscopy (cryo-EM) grids has further increased the efficiency of data collection[16–18]. Similar to developments in single-particle cryo-EM[19–23] these advances have motivated the development of strategies for on-the-fly data pre-processing that can provide real-time feedback on image quality[24–27];

[1]Department of Biochemistry, Duke University, Durham, NC, USA. [2]Department of Computer Science, Duke University, Durham, NC, USA. [3]Department of Electrical and Computer Engineering, Duke University, Durham, NC, USA. [4]Department of Chemical Engineering, Massachusetts Institute of Technology, Cambridge, MA, USA. [5]These authors contributed equally: Hsuan-Fu Liu, Ye Zhou. ✉e-mail: alberto.bartesaghi@duke.edu

however, adoption of these strategies has been slow, due to the increased complexity, limited robustness and higher computational demands of routines for tilt-series alignment, tomogram reconstruction, contrast transfer function (CTF) estimation and particle picking.

Particle picking in particular, poses major challenges due to the heterogeneous and crowded nature of native cellular environments and the distortions caused by the missing wedge[28]. Nonetheless, several strategies for three-dimensional (3D) particle picking have been proposed. Template matching can be used when an external reference is available, but is computationally expensive and may introduce model bias[29]. To detect membrane proteins attached to viruses, multi-step strategies that rely on 3D segmentation can be used, but these require manual user intervention making them unsuitable for high-throughput tomography[30,31]. More recently, neural network-based approaches have been introduced, but these use fully supervised models that require extensive labeling, long training times and often need retraining when applied to new datasets[32,33].

During 3D refinement, conventional sub-tomogram averaging (STA) relies on the extraction of sub-volumes from large tomograms (up to several GB in size). For datasets with hundreds of tilt series and tens of thousands of particles, this strategy results in prohibitively large storage and computational requirements that make image analysis impractical. In contrast, strategies for constrained SP-CET bypass the need to extract sub-tomograms and only use two-dimensional (2D) projections during refinement, resulting in substantial storage savings[34]. In practice, however, even this solution may require unreasonable amounts of space and create input/output (I/O) bottlenecks during downstream processing. Moreover, frameworks based on constrained SP-CET still require the generation of sub-tomograms for ab initio pose assignment and 3D classification[35–38]; a strategy that scales poorly for the analysis of large datasets needed for in situ structural studies.

Existing frameworks for SP-CET focus on particular aspects of the data analysis pipeline, but the combination of different processing steps is still hindered by the lack of unified data workflows[39]. Some tools are used for data pre-processing[26,27] and others are used for tilt-series alignment and reconstruction[24,25], CTF estimation[40], particle picking[30,38] and 3D refinement and classification[35–38]. A comprehensive platform for streamlined processing of all steps in the SP-CET pipeline, along with an interactive user interface, is lacking (Supplementary Table 1).

Here, we present an end-to-end framework for SP-CET data processing that exclusively uses 2D raw data for all downstream 3D refinement and classification tasks. Avoiding sub-tomogram extraction and saving of particle stacks, results in a scalable strategy that can be used to process datasets with hundreds of thousands of particles. To support high-throughput workflows, we present routines for automated pre-processing of tilt series, including techniques for unsupervised and semi-supervised particle picking. For refinement and reconstruction, we propose algorithms for reference-based alignment, global and region-based constrained refinement, particle-based CTF determination, beam-induced motion correction and exposure weighting and 3D classification for variability analysis. We validate our framework on benchmark datasets from in vitro and cellular samples, demonstrating its ability to extract high-resolution information and characterize conformational heterogeneity of native proteins. These tools are implemented in the package nextPYP, a fully featured application with an easy-to-use graphical user interface (GUI).

## Results

### On-the-fly processing of tilt series

Monitoring data quality during high-throughput acquisition (Fig. 1a) is especially important in tomography where the unit of data is a series of projections acquired from the same area of the sample. This imposes stringent tracking and defocus requirements that must be met to ensure the integrity of the data. We developed robust routines for on-the-fly processing of tilt series that can produce particles and metadata ready for SP-CET refinement. Our data pre-processing strategy includes the

following steps: (1) gain correction and video frame alignment; (2) alignment of tilt series using fiducial-based or patch-tracking methods; (3) tilted CTF estimation; (4) tomogram reconstruction; and (5) particle picking (Fig. 1b). Tilt series are processed in parallel on a multi-core server or computer cluster and the results are live-streamed to a GUI (Extended Data Fig. 1). The tool is designed to support the analysis of hundreds of tilt series and produce the necessary metadata for subsequent high-resolution and conformational variability analysis (Fig. 1c). Typical running times are reported in Supplementary Table 2.

### Size-based particle picking

Size-based strategies have been used effectively to pick particles from monodisperse samples in single-particle cryo-EM. Tomography samples are obviously not monodisperse, but these techniques are still useful and can facilitate particle picking from in vitro samples used for benchmarking or large complexes such as ribosomes imaged in situ. We propose a size-based particle detection approach based on our previous method for 2D particle picking[41]. Tomograms are first low-pass filtered to highlight features in the desired size range, followed by local minima detection and removal of weak peaks. To avoid picking particles from areas near gold-fiducials or contamination, a pre-processing step is applied to automatically detect these regions. This approach only requires specification of the particle size and runs in tens of seconds.

### Membrane-constrained template matching

To handle a wider range of tomography samples, nextPYP also implements a three-step strategy to find native membrane proteins attached to the surface of viruses or liposomes. First, we automatically determine the location of virions within tomograms using a Hough transform voting scheme, followed by segmentation of each virion in 3D using density-driven minimal surfaces[42]. During a third step, we use a low-resolution map as a reference to run constrained template matching where the search is restricted to the area outside the segmented membrane. Using this procedure, both the location of membrane-bound particles and their normal orientations can be obtained, which can be used to speed up orientation search during refinement. The algorithm to locate virions only requires specifying the average radius in Å, whereas the segmentation algorithm detects arbitrarily shaped membranes contained between two concentric spheres of a user-defined radius around the virion centers. If a reference for template matching is not available, it is possible to pick uniformly spaced positions along the segmented membranes.

### Semi-supervised particle picking using deep neural networks

We also implemented a semi-supervised particle-picking strategy using deep neural networks. Unlike fully supervised approaches that require densely annotated images or tomograms for training[32,33,43], our semi-supervised networks use multi-task learning, data augmentation and consistency regularization to overcome the need for labor-intensive labeling[44,45]. As a result, these routines require only a sparse set of annotations (several particles from a few tomograms) and can be trained in minutes rather than hours. nextPYP implements a convenient GUI for interactive labeling of particles in three dimensions, training the deep-learning models, running inference on hundreds of tomograms and evaluating the particle picking results in three dimensions.

### A scalable framework for structure determination by SP-CET

To reduce the storage footprint of traditional STA workflows (Fig. 2a), nextPYP bypasses the generation of large tomograms, sub-volumes and the saving of particle stacks by reading the data directly from the unprocessed, unaligned raw tilt series images, Fig. 2b. This approach results in storage savings of several TB per dataset, ensures the efficient utilization of resources, simplifies data management and allows processing of larger datasets than previously possible. Moreover, the removal of I/O bottlenecks associated with generating, saving and transferring

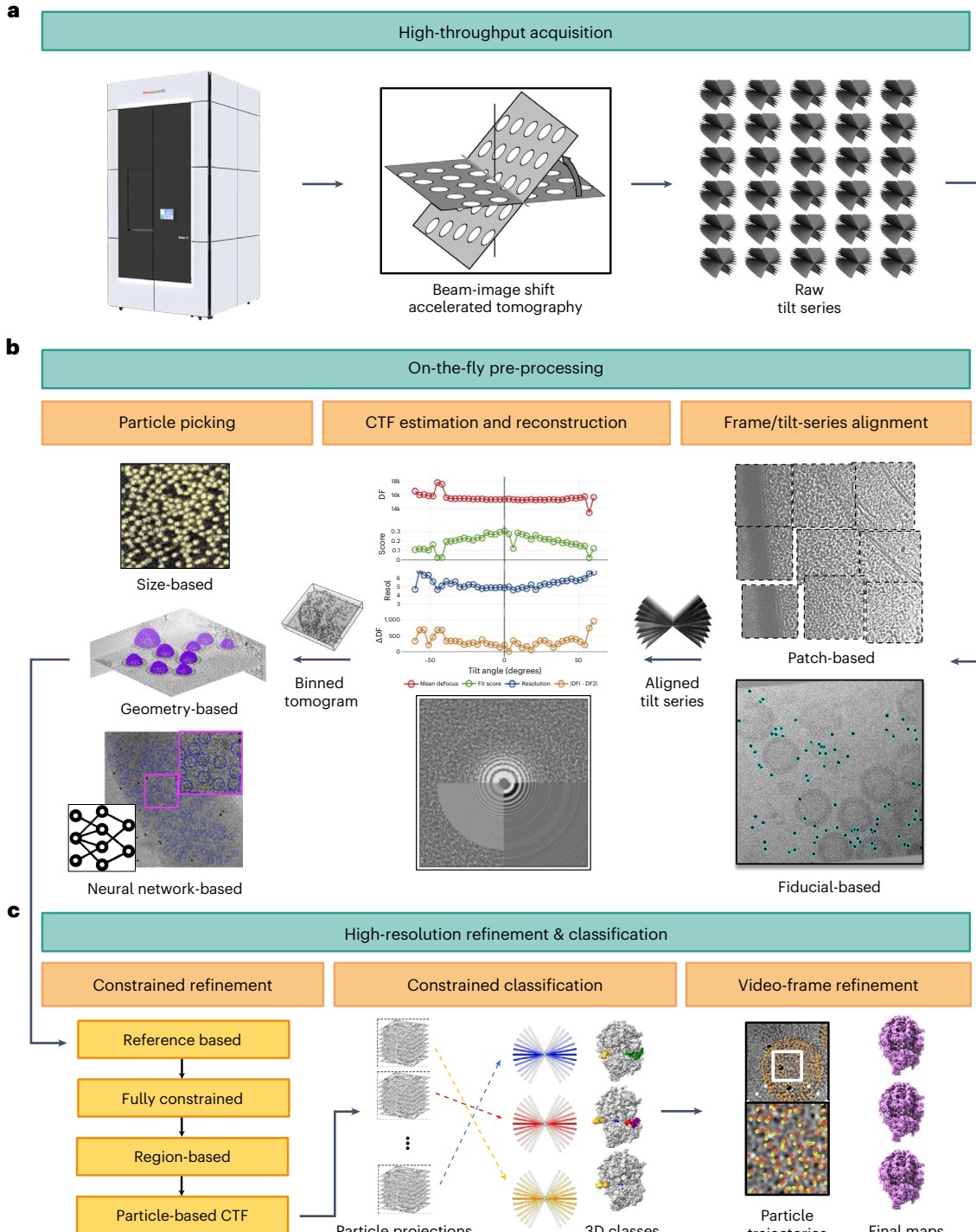

**Fig. 1 | Comprehensive end-to-end pipeline for single-particle cryo-electron tomography.** Sequence of steps required to convert raw tilt series into high-resolution structures. **a**, High-throughput data collection using beam-image shift accelerated cryo-electron tomography produces raw tilt series in parallel. **b**, Pre-processing of tilt series is performed on-the-fly during data collection and includes the steps of frame and tilt-series alignment (fiducial-based and patch-based using equal size patches), tilted CTF estimation and reconstruction and particle picking using size-based, geometry-based or neural network-based approaches. DF, defocus; Resol, resolution. **c**, High-resolution constrained refinement consists of reference based, fully constrained, region-based and particle-based CTF refinement steps, followed by constrained 3D classification and video frame refinement resulting in final high-resolution maps.

large image stacks results in faster processing, allowing the routine analysis of hundreds of thousands of particles. Tilt series are processed in bundles and only one intermediate 3D reconstruction per bundle is saved to permanent storage (typically just a few hundred MB). These reconstructions are then merged into a final 3D map and this process is iterated multiple times. The modularity of this architecture makes it well suited for use in cluster environments where frequent access to network file systems can introduce serious performance bottlenecks.

### Fully constrained refinement of particle poses
Our original constrained SP-CET work addressed the issue of overfitting by imposing constraints exclusively on the orientations assigned to

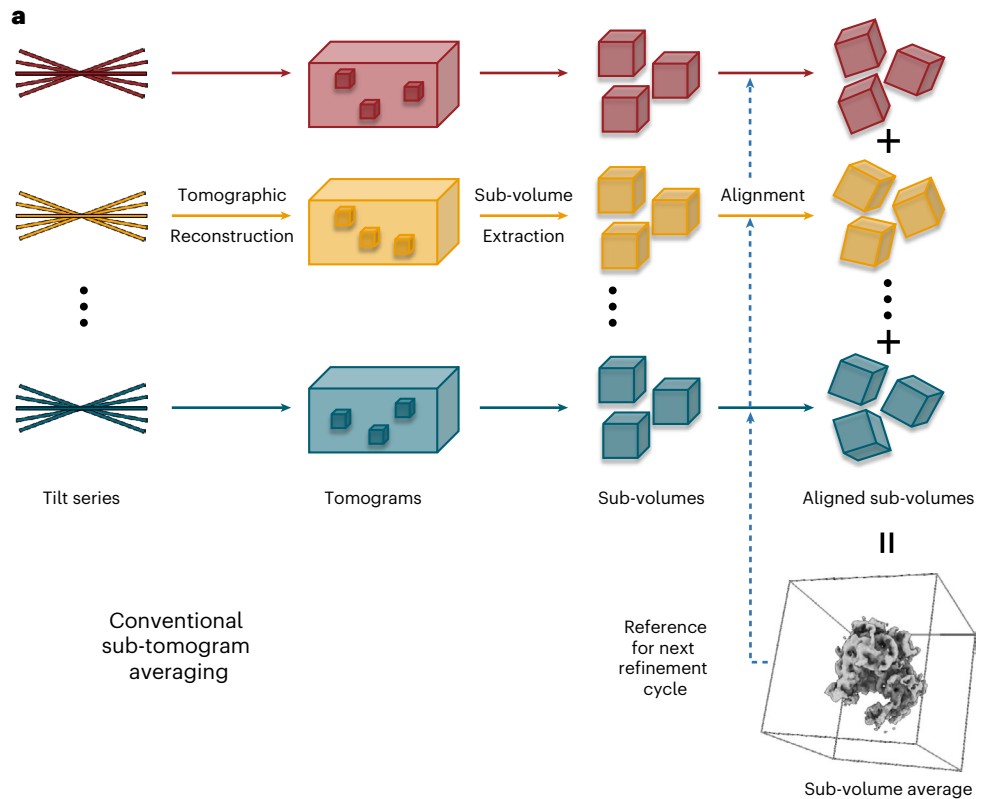

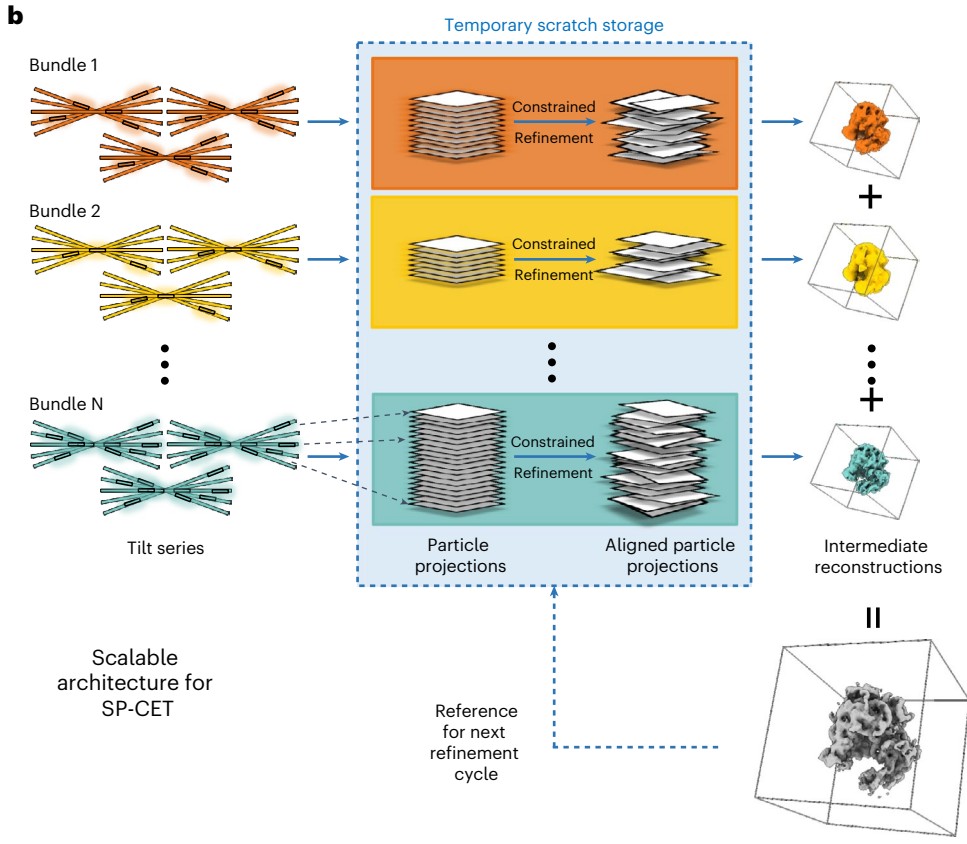

**Fig. 2 | Scalable architecture for in situ structure determination using SP-CET.** Scalability to large datasets is achieved by avoiding the extraction and saving of sub-tomograms and large particle stacks. **a**, Schematic of traditional approach for STA where data storage and I/O can become serious bottlenecks. **b**, Schematic of scalable framework for SP-CET refinement that only requires storage of raw data and final reconstructions. Bundles of tilt series are processed in parallel tracks using multiple cores or servers. Particle projections are extracted and aligned on-the-fly using temporary scratch space and partial reconstructions from each track are merged into a final map.

each particle projection[34]. In this case, there are three rotation angles per particle and two translations per particle projection that need to be determined during alignment. By ensuring that the constraints of the tilt geometry were satisfied throughout refinement, this strategy resulted in higher resolution reconstructions and minimized overfitting. For tilt series with lower defocus or from crowded cellular samples; however, the weaker signal in each particle projection can still lead to overfitting even when the rotational constraints of the tilt geometry are imposed. To address this problem, we implemented a fully constrained algorithm where both rotations and translations of the tilt geometry are enforced throughout refinement. This strategy has the advantage of reducing the number of degrees of freedom to six parameters per particle (three rotation angles and three translations), making it more robust to overfitting and extending its applicability to a broader set of cryo-ET samples, including lower defocus tilt series and datasets from cellular environments.

## Reference-based pose estimation directly from 2D projections

Existing strategies for constrained SP-CET refinement require access to pre-existing particle poses so they can later be refined using the 2D projection data. The initial poses are typically obtained using traditional STA, which in turn requires extraction and alignment of sub-volumes. Even when sub-volumes are extracted using large pixel sizes (to reduce space), this adds an unnecessary step that requires additional compute and storage resources and the use of different software packages which can introduce compatibility issues. Here, we propose a reference-based approach that directly uses 2D particle projections to derive initial poses for every particle. To determine alignment parameters for each particle projection, we use a fully constrained global search approach where each particle is assigned the sum of scores from all its tilted projections (the allowable range of rotations and translations with respect to the original particle angles and positions can be controlled by the user). To speed up the search, we use binned images and a coarse grid to sample the three Euler angles. As this approach only uses the raw 2D data, it is computationally efficient, is not affected by the missing wedge and simplifies the handling of metadata for the refinement steps downstream.

## Region-based constrained projection matching

Our fully constrained framework prevents noise overfitting during particle orientation determination by considering all tilted projections of a particle as a rigid body and using the signal from all particles in the field of view to refine the parameters of the tilt geometry. This approach, however, does not consider the local distortions that may occur due to beam-induced motion and deformation of the sample during data acquisition. To account for these effects, we developed a region-based constrained refinement strategy that refines the parameters of the tilt geometry using subsets of particles present within regions of a tomogram. Similar to M[37], this strategy relaxes the global constraints of the tilt geometry allowing the estimation of local deformations. The number of regions is determined based on the amount of signal present in each sub-area (typically proportional to the molecular weight and concentration of particles) and the extent of deformation expected in the sample. Provided that there are sufficient particles within each region, this strategy can be effective at preventing overfitting while leading to substantial improvements in map resolution.

## Particle-based CTF refinement

In the constrained SP-CET framework, CTF parameters are assigned to individual particle projections using the defocus parameters estimated for each image in a tilt series and the 3D position of particles within tomograms[13]. This process, however, is not always accurate and can lead to imprecise assignment of defocus parameters that may ultimately limit resolution. Similar to per-particle CTF refinement strategies first implemented for single-particle cryo-EM in FREALIGN[46], we use

the most recent 3D reconstruction and our region-based constrained refinement approach to locally refine defocus and astigmatism for images within each region. CTF parameters for each particle projection are then re-calculated, leading to measurable improvements in map resolution.

## Video frame refinement and self-tuning exposure weighting

Modern electron detectors record each tilted image as a sequence of video frames. Similar to strategies used in single-particle cryo-EM, the time-resolved data can be used for tracking the movement of individual particles through the exposure to correct for beam-induced motion. Unlike single-particle cryo-EM; however, the dose accumulated in each frame is too low to allow accurate alignment[47]. To overcome this, we use running frame averages and produce noisy particle trajectories that are later regularized using spatial and time constraints[48] (Fig. 3a). This process generates a similarity score for each frame (Fig. 3b), which can be used to perform frequency-dependent exposure weighting (Fig. 3c). The weights we obtained indicate that only frames from the first four low-tilt exposures contribute high-resolution information to the reconstruction. Despite the low signal-to-noise ratio (SNR), the curve for each tilt shows a similar characteristic bell-shape as previously reported for single-particle cryo-EM[48]. The reduction in average score with each new tilt is consistent with the onset of radiation damage and thickening of the specimen due to tilting. To get a cleaner picture of the score variations within an exposure, we subtracted the mean score for each tilt from the corresponding per-frame scores and averaged the resulting curves from all the tilts (Fig. 3d). While video frames for tilt series were already used by M for motion correction[37], the frames were averaged before reconstruction to reduce storage and compute requirements. Using our scalable architecture for SP-CET refinement, we can calculate exposure-weighted maps directly from the video frames, thus providing increased efficiency during the extraction of high-resolution signal.

## Validation using HIV-1 Gag benchmark dataset

We validated nextPYP on tilt series from the HIV-1 Gag dataset from EMPIAR-10164 (ref. 49). Tilt series in super-resolution mode were first pre-processed using our routines for on-the-fly data analysis. The position of HIV-1 virions was automatically detected, followed by 3D segmentation and selection of particle locations from their surface. The normals at each position were used to restrict the reference-based search to a single in-plane angle. Fully constrained refinement, region-based refinement (using a 8 × 8 × 2 grid), particle-based CTF refinement, video frame refinement and additional rounds of region-based refinement, resulted in a 3.2 Å resolution map using 14,482 particles extracted from five tilt series (Fig. 4a and Extended Data Fig. 2a,b). All refinement operations were performed using particles extracted in 2× binned precision (1.35 Å per pixel) and a box size of 384 pixels. Only tilts in a ±6° range were used for refinement (as these were the ones with the highest contribution to high-resolution), whereas the entire ±60° range of projections was used for 3D reconstruction. We also analyzed the entire set of 43 tilt series using the same strategy and obtained a 3.0 Å map from 109,496 particles, matching the resolutions obtained by M[37] and Relion[36], while using 16% and 24% fewer particles, respectively (Extended Data Fig. 2c,d). Our results were obtained without extracting sub-volumes or saving image stacks, resulting in ~25 TB of storage savings and substantially faster processing (timing and data processing details are included in Supplementary Tables 2 and 3, respectively).

## Sub-2 Å structure of mouse heavy-chain apoferritin

We also validated nextPYP using tilt series of apoferritin downloaded from EMPIAR-11273. One hundred raw tilt series were pre-processed using patch-based tracking for tilt series alignment. Particles were detected automatically using our size-based particle picker using a

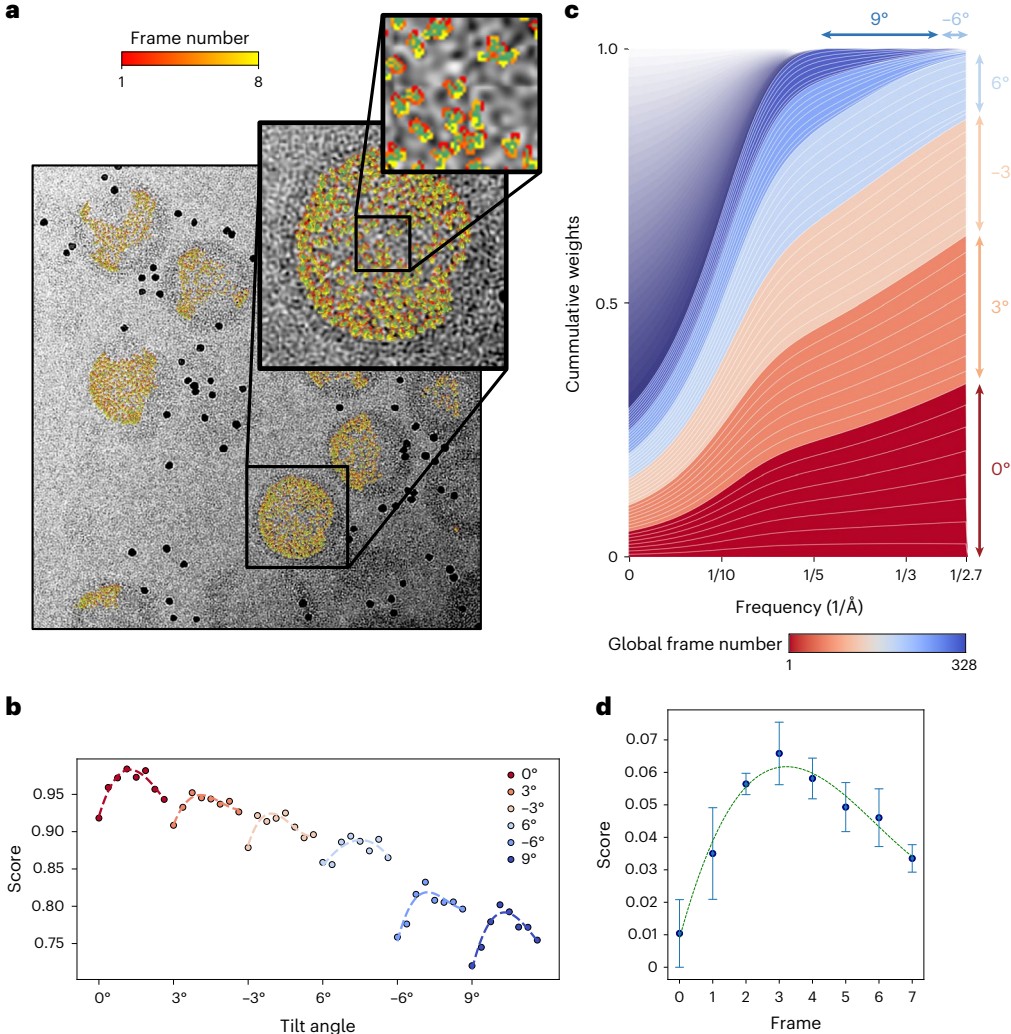

**Fig. 3 | Beam-induced motion and exposure weighting of tilt-series data.**
**a**, 0-tilt image from HIV-1 Gag tilt series (TS_03) showing movement trajectories for individual particles estimated using video frame alignment routines (similar trajectories were obtained for the remaining 42 tilt series from EMPIAR-10164). **b**, Average score distribution of individual particle frames for tilt images at 0°, 3°, −3°, 6°, −6° and 9° showing relative differences in image quality between exposures. **c**, 2D score-based weights used during 3D reconstruction for each frame as a function of spatial frequency. **d**, Average per-frame score distribution over full dataset. Data are presented as mean values ±s.d. calculated over $n = 6$ video frames corresponding to the first six exposures in the tilt series (0°, 3°, −3°, 6°, −6° and 9°).

radius of 75 Å. Reference-based search using 2D data was followed by several rounds of fully constrained and region-based refinement (using a 4 × 4 × 2 grid), particle-based CTF refinement, video frame refinement and additional rounds of region-based refinement, resulting in a 1.8 Å resolution map (Fig. 4b).

**Constrained classification for characterizing heterogeneity**
Unlike conventional sub-tomogram classification where volumes are clustered based on their 3D structural features[50,51], we propose a strategy to analyze conformational heterogeneity that exclusively uses the raw 2D particle projection data (Extended Data Fig. 3). Similar to the original constrained SP-CET approach[34], this has several advantages: (1) it requires fewer compute and storage resources; (2) is not affected by the effects of the missing wedge; and (3) can leverage existing classification tools for single-particle cryo-EM. Our strategy is based on the 3D classification algorithm implemented in cisTEM[52], with additional components required to support our distributed processing architecture and the imposition of the tilt geometry constraints. To ensure that all projections from a given particle are assigned to the same class, per-particle occupancies are calculated by averaging the occupancies assigned to individual projections, Extended Data Fig. 4. Constrained classification can be run simultaneously with global or region-based refinement, or without alignment. The latter option is useful when the signal is too weak to allow reliable orientation assignment and can help prevent overfitting by limiting the dimension of the search space. The computational efficiency and small storage footprint of our classification strategy allows us to routinely analyze hundreds of thousands of particles (Supplementary Table 3).

**Disentangling translation states from in vitro 80S ribosomes**
We validated our classification approach on tilt series from in vitro 80S ribosomes[53] (EMPIAR-10064, defocus data). From a total of ~4,000 particles, we first obtained a consensus map at 5.6-Å resolution as reported previously[13], followed by 30 iterations of 3D classification into five classes representing different translational states of the 80S ribosome (Fig. 5a). Based on the presence of cofactors and differences in the rotation of the 40S subunit (Fig. 5b), we assigned the classes into two subgroups. Classes 1 and 2 show a translocation direction rotating 40S and clear density of eEF2, which correlates highly with the translocation state. Classes 3, 4 and 5 show P-site transfer RNA, E-site tRNA and the

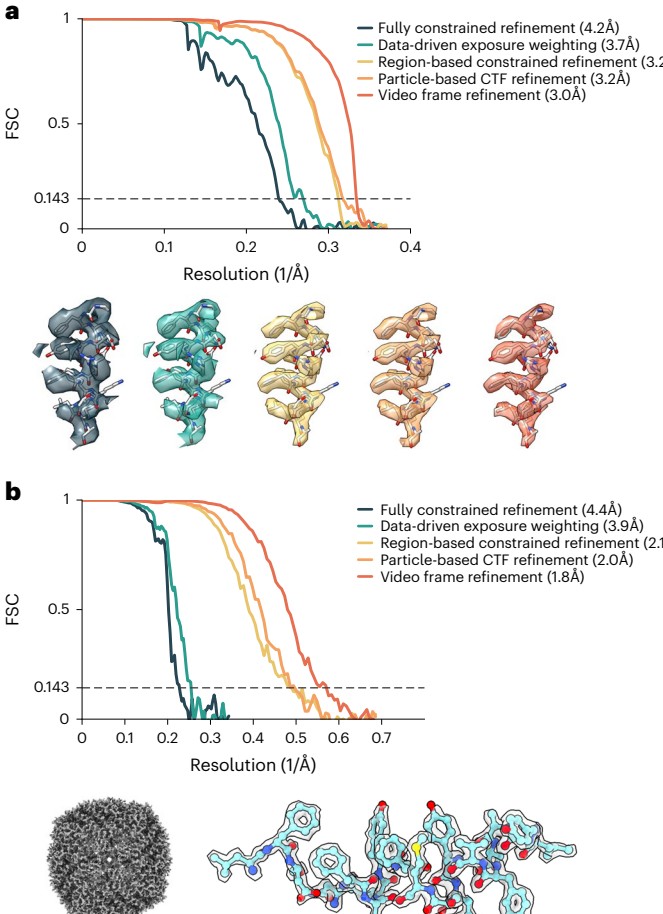

**Fig. 4 | Resolution improvement and validation using in vitro samples of HIV-1 Gag and apoferritin. a**, Resolution improvements obtained by nextPYP using 43 tilt series from EMPIAR-10164 after fully constrained refinement (4.2 Å), dose-weighted reconstruction (3.7 Å), region-based refinement (3.2 Å), particle-based CTF refinement (3.2 Å) and video frame refinement (3.0 Å). Corresponding Fourier shell correlation (FSC) curves calculated between half maps (0.143 cutoff criteria was used to determine resolution) (top). Alpha helical segment with model fit into map showing the progression in resolution with each step. **b**, High-resolution structure of apoferritin obtained using 31,890 particles extracted from 100 tilt series (EMPIAR-11273). FSC plots between half maps showing incremental improvements in resolution from each processing step (top). The 1.8 Å resolution reconstruction and selected high-resolution features with atomic model fit into the density (bottom).

eEF1A–A/T-tRNA complex, as well as the canonical 40S conformation characterizing the post-translational state. Classes in subgroup 1 share the same cofactors, but are differentiated by distinct L1 stalk conformations (class 1 with L1 in closed position and class 2 with L1 in open position). More detailed comparison of classes 1 and 2 revealed minor rotation differences between the 40S subunits. Rigid body fitting into the maps using PDB ID 6GZ3 and PDB ID 6GZ4 coordinates confirmed that class 1 is in the translocation-intermediate-POST-1 state and class 2 is in the translocation-intermediate-POST-2 state. We hypothesize that the missing tRNA in subgroup 1 may be the consequence of protein purification. All classes in subgroup 2 show good agreement with the translation elongation complex (PDB ID 5LZS), which is in the decoding state. E-site tRNA is released in class 3 with the L1 stalk present in an open conformation. Class 4 is in the pre-loading state without the eEF1A–tRNA complex, whereas E-site tRNA is pre-released and L1 is in the intermediate state. The eEF1A–tRNA complex and L1 stalk-attached E-site tRNA coexist in class 5. Release of the E-site tRNA is not strictly

coupled to binding of aminoacyl-tRNA in the A-site, which is consistent with observing evenly distributed classes in subgroup 2. We also compared our results to previous classification methods implemented in the package emClarity[35] (Fig. 5c,d and Supplementary Fig. 1a). In summary, nextPYP results show better map quality and clear density for cofactors, allowing unambiguous assignment of ribosome translation states.

## Heterogeneity analysis of ribosomes from *M. pneumoniae* cells

Tilt series collected from intact *Mycoplasma pneumoniae* cells treated with chloramphenicol were downloaded from EMPIAR-10499 (ref. 37). Starting from ~18,000 particles picked using our semi-supervised deep-learning approach, we performed reference-based alignment to obtain a consensus map at 3.9-Å resolution (Extended Data Fig. 5). This was followed by 45 iterations of constrained classification into three classes (Fig. 6a and Supplementary Fig. 1b). The chloramphenicol-treated cells only showed decoding and peptidyl transfer (classic non-rotate) states, which is consistent with the antibiotic function. A small population (28%, class 1) was still in the decoding state, right before releasing the elongation factor EF-Tu protein. In addition, ~70% of the particles (classes 2 and 3) were stalled at the peptidyl transfer state characterized by a 4° rotation of the 30S subunit (Fig. 6b). The presence of the inhibitor provided a longer time window for E-site tRNA to be released, resulting in missing density for E-site tRNA in all classes. Classes 2 and 3 showed a 6° swing movement of the L1 stalk (Fig. 6c). Together, these results demonstrate the ability of our approach to study protein conformational variability in situ.

## Translating 80S ribosome conformations from in situ lamellae

Finally, we benchmarked our methods on tilt series obtained from cryo-FIB-milled lamellae from a cellular sample containing 80S ribosomes[14] (EMPIAR-10987). A total of 6,763 particles were picked using our deep-learning approach from 20 tilt series and used to obtain a consensus map at 8.4-Å resolution (Supplementary Fig. 1c). We calculated average scores per tilt and confirmed that the relative contribution of each tilt was consistent with the accumulation of dose and the change in tilt angle (Extended Data Fig. 6). We then executed 30 iterations of 3D classification using a focus mask centered at the P-site tRNA position and a radius of 110 Å. After multiple classification runs using different numbers of classes, the presence of five stable conformations was detected (Fig. 6d). Classes were resolved at around 11–12-Å resolution and cofactors associated with different conformations could be unambiguously recognized (Supplementary Fig. 1d). Classes 1, 2 and 3 were in a similar non-rotated state. Classes 1 and 3 showed density for the canonical P-site tRNA, but different L1 stalk conformation (class 1 has the L1 stalk in a closed position, whereas class 3 has the L1 stalk in an open position) and class 2 containing ~700 particles showed no density for the P-site tRNA. These three classes accounted for ~40% of the ribosome population and can be assigned to the recycling/initialization state. Class 4 was also in the non-rotated state, with the 40S rotated by the largest angle (in the backward direction; Fig. 6e) and eEF1A–A/T-tRNA bound, which was in the pre-peptide bond formation decoding state, accounting for 30% of the particles. Class 5 was in the rotated state with translation elongation factor eEF2 bound, the 40S body was rotated by 8° and the 40S head was rotated by 17° and accounts for another 30% of the particles. Alignment with different translocating coordinates, confirmed that class 5 was in the translocation-intermediate-POST-3 state. No density for E-site tRNA was identified in any of the classes, which could be the result of the limited number of particles. Overall, these results demonstrate the ability of our constrained classification approach to successfully separate translating conformations of ribosomes from cryo-FIB-milled lamellae.

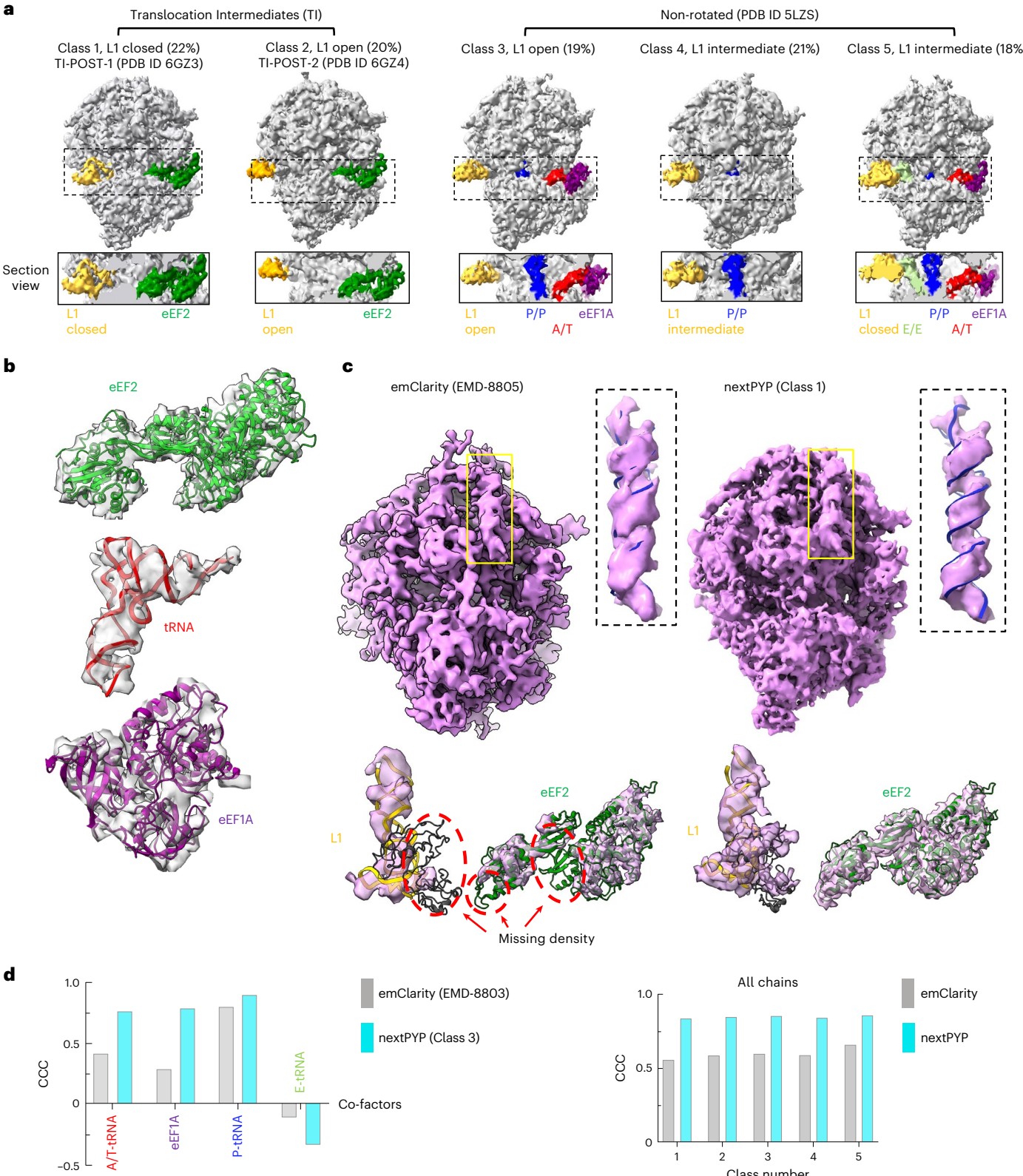

**Fig. 5 | Constrained classification of mammalian 80S ribosomes (EMPIAR-10064). a**, Five classes of translating 80S ribosomes bound to different cofactors. Class 1 and 2 are in pre-translocational state with rotating 40S. L1 stalk is colored yellow, eEF2 is colored green. Class 3, 4 and 5 are in the post-translocational state with non-rotating 40S. eEF1A is colored magenta, A/T-tRNA is colored red, P-site tRNA is colored blue and E-site tRNA is colored green. Section views show the different cofactors inside the inter-subunit cavities. **b**, Representative density for eEF2, tRNA and eEF1A with corresponding coordinates fitted into the density. **c**, Comparison of map quality for class 1 between emClarity and nextPYP. Insets show comparison of density for helical RNA. Coordinates were fitted into the density for cofactor eEF2 (green) and L1 stalk (yellow) for both maps (shown at the bottom). **d**, Histogram of TEMpy2 (https://tempy.ismb.lon.ac.uk/) model-to-map CCC scores for nextPYP and emClarity reconstructions (left, cofactors in class 3; right, all chains from all five classes). Positive scores represent better fits and negative scores indicate that no corresponding cofactor density was present.

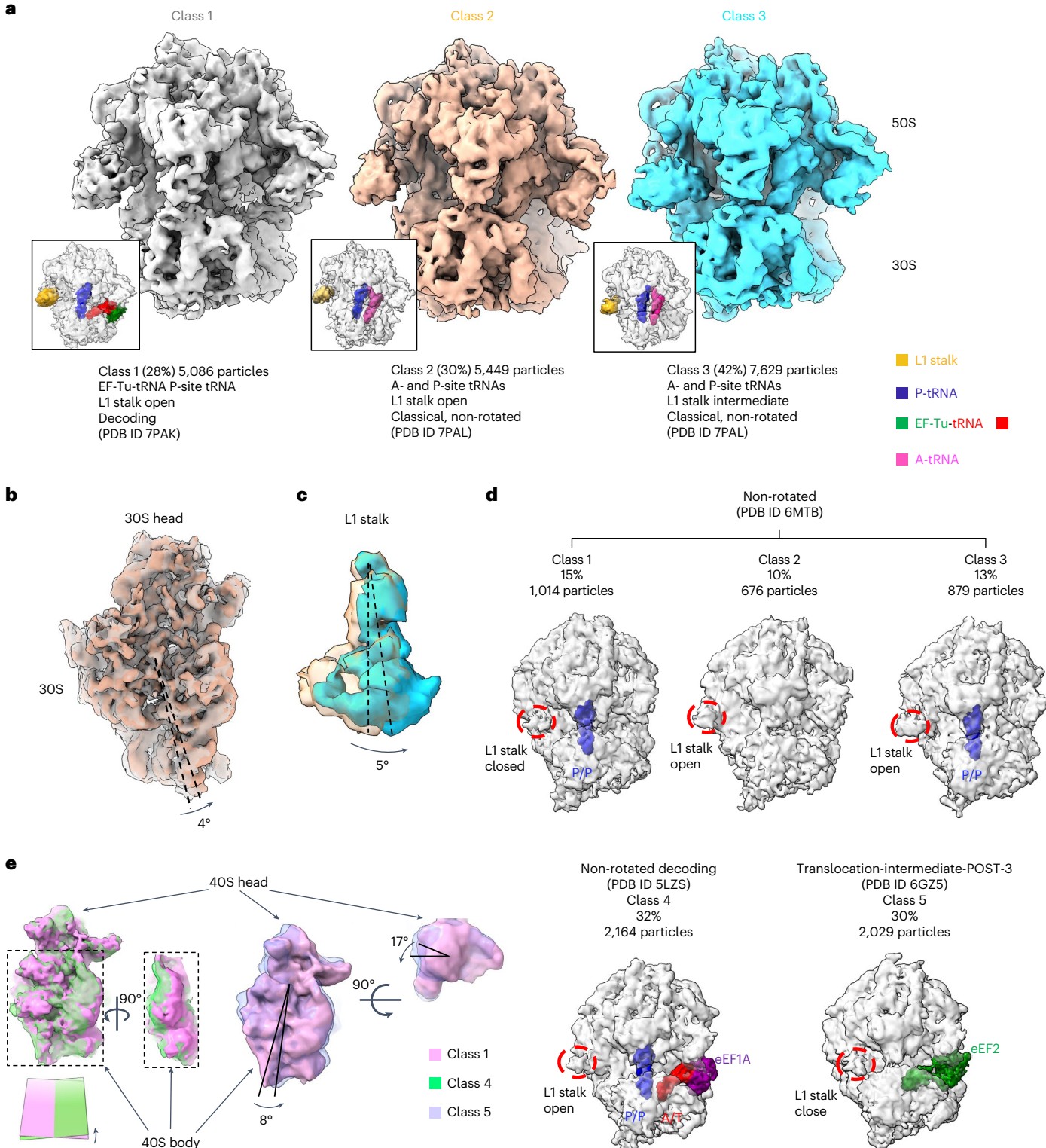

**Fig. 6 | Classification of 70S ribosomes from intact cells and 80S ribosomes from cryo-FIB lamellae. a**, Three classes of 70S ribosomes with different translating states or L1 stalk conformations. Class 1 is in decoding state with EF-Tu binding tRNA adopting to A-site. Class 2 and 3 are in classic non-rotated state, showing A-site and P-site tRNAs, while L1 stalks are in different positions (class 2, L1 stalk is open; class 3, L1 stalk is intermediate close, highlighted in **c**). **b**, Superimposed maps for class 1 and 2 viewed from the bottom of the 30S subunit showing the small subunit rotation during tRNA loading. **c**, L1 stalk superimposed density for class 2 and 3 showing the open and half-closed states. **d**, Classes of 80S ribosomes from cryo-FIB lamellae. Classes 1, 2 and 3 are in non-rotated states, while class 1 and class 3 have binding P-site tRNA differentiated by close and open L1 stalk states and class 2 with no tRNA bound. Class 4 is in the decoding state with cofactor eEF1A and A/T-site tRNA, P-site tRNA bound. Class 5 is in the post-translational state with cofactor eEF2. **e**, Superimposed maps of 80S class 1 and 2 showing rotating subunits, with plates representing the 40S body central planes.

## Discussion

Structural studies using single-particle microscopy require imaging and averaging many copies of the target of interest to remove noise and improve resolution. Unlike in vitro studies that use purified protein at high concentrations, proteins imaged within their native cellular context are present at much lower physiological concentrations, requiring imaging of large areas of the sample to accumulate enough copies to achieve high-resolution. Advances in sample preparation, specimen screening and high-speed tomography, have increased the throughput of data collection making it possible to acquire hundreds of tilt series during a single microscopy session. Strategies for SP-CET data analysis; however, have lagged behind and currently require extensive user input and access to prohibitively large computational and storage resources, constituting a major obstacle for the development and broad adoption of this powerful technique.

Here, we present nextPYP, a scalable framework for SP-CET data analysis that can routinely convert hundreds of tilt series or hundreds of thousands of particles into high-resolution structures. The software uses a combination of re-implementations of existing techniques and new methods, as summarized in Supplementary Table 4. Bypassing the need to generate sub-tomograms and particle stacks results in a light-weight architecture with a small storage footprint that is scalable and substantially accelerates data analysis. Our robust strategies for on-the-fly pre-processing of tilt series allow monitoring of data quality during acquisition and seamless conversion of metadata for subsequent 3D analysis. We present efficient algorithms for reference-based alignment and high-resolution refinement, including region-based and particle-based CTF refinement, video frame alignment and data-driven exposure weighting. Our strategies for 3D classification use the constraints of the tilt geometry to effectively disentangle structural variability of flexible complexes imaged in situ. We validate our approach using in vitro and cellular datasets demonstrating its ability to achieve high resolution and uncover translating states of native ribosomes. These tools are available in the software package nextPYP, a scalable and easy-to-use platform for SP-CET data analysis designed to facilitate adoption of this technique by the structural biology community.

nextPYP successfully overcomes fundamental bottlenecks in the SP-CET data analysis pipeline; however, most datasets still require access to multi-node computer clusters or scalable cloud-based solutions. In particular, reference-based 3D reconstruction requires exhaustive sampling of poses between each particle and the reference, resulting in low-throughput performance. Future work will be needed to further reduce the computational footprint of constrained projection alignment and to design efficient search strategies for robust ab initio reconstruction. Together with recent developments in specimen preparation and data collection, these advances will help accelerate the structural analysis of native samples and contribute to advancing the field of visual proteomics.

## Online content

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

## Methods

### On-the-fly pre-processing of tilt series

Gain correction, video frame alignment and averaging were performed using a modified version of the program unblur[54] designed to minimize I/O operations and speed up processing. Tilt-series alignment was performed using routines implemented in IMOD and include protocols for fiducial-based and patch-based alignment using non-overlapping and uniformly sized patches[55]. Per-tilt CTF was estimated using the procedure we reported previously[13] (based on CTFFIND4[40]) and only binned tomograms were produced for the purpose of assessing reconstruction quality using IMOD's tilt command. Images in a tilt series were processed in parallel using multiple CPU threads. Steps such as tomogram reconstruction and CTF estimation that do not depend on each other were also parallelized. Information was live-streamed to the GUI, providing convenient access to data quality metrics and assessment of tomograms.

### Strategies for 3D particle picking

**Size-based particle picking.** Particle picking was conducted using the binned tomograms produced during data pre-processing. A robust contamination-detection step was executed as follows: (1) we applied a high-pass filter to the original tomogram and binarized the result (using the mean density plus 3 × s.d.) to detect high-contrast regions; (2) we removed small objects from this mask and filled in small holes using mathematical morphology operations; and (3) we dilated the mask to prevent picking particles that were too close to contaminated areas. Tomograms were then low-pass filtered to highlight features within the user-specified size range and particles were detected at the positions of local minima of the density that fell outside the contamination mask. To reduce the effects of the missing wedge, we erased the gold from the raw tilt series before reconstruction using protocols implemented in IMOD[24].

**Geometry-based picking.** Virions were detected based on a robust voting mechanism that uses the 3D Hough transform implemented in ITK (https://itk.org/) and VTK (https://vtk.org). This approach produces the center coordinates and an estimated value for the radius of each virion. This information was used to generate separate tomograms for each virion (using a smaller pixel size than that used to generate the tomograms during pre-processing), followed by 3D segmentation using minimal-surfaces algorithms[42]. An external reference was then used to execute template search constrained to the outside of the segmented surfaces. Virions were processed in parallel to reduce running times.

**Semi-supervised picking based on deep neural networks.** We used sparse labels selected in three dimensions to train a semi-supervised deep neural network as described elsewhere[45]. We manually selected several particles in three dimensions from a few tomograms using nextPYP's built-in visualization tools. The model was first trained using a GPU, followed by inference on all tomograms in a dataset. Particle picking results were visually inspected for accuracy. For challenging datasets, the number of annotations can be increased and used for training until the desired accuracy is achieved. For validation, we applied this approach to pick 35,352 particles of native enzyme ribulose-1,5-biphosphase carboxylase-oxygenase (RuBisCO) from *Chlamydomonas reinhardtii* cells[56] and used them to produce a 12-Å resolution map (Extended Data Fig. 7).

**Comparison between size-based and neural network-based picking.** We compared the performance between these approaches using in vitro and in situ datasets, EMPIAR-10304 and EMPIAR-10499, respectively. Performance in terms of precision, recall and F1 was slightly higher for the neural network approach, but the size-based approach was faster because it did not need manual labeling or training (Extended Data Fig. 8 and Supplementary Table 5).

### SP-CET data analysis without sub-volumes and particle stacks

In conventional STA, storing 12,000 sub-volumes using a box size of 384 requires 2.5 TB of space. Applying the original constrained SP-CET approach to the same dataset (assuming tilt series were acquired using a standard ±60 tilting scheme with 41 projections), requires generating an image stack with almost half a million particle projections (12,000 × 41 = 492,000 images). Using a box size of 384 voxels, such a stack will still require 270 GB of disk storage, creating substantial I/O bottlenecks for the processing downstream. Instead, nextPYP bypasses the generation of sub-volumes and saving of large particle stacks by leveraging its parallel computing architecture and processing tilt series in discrete bundles (Fig. 2b). Internally, the program still produces 2D particle stacks because they are needed for running refinement and reconstruction in cisTEM[52], but these are only stored temporarily on local scratch space, which typically consists of fast SSD drives. When running on high-performance computing clusters, this strategy minimizes network traffic, resulting in faster processing speeds and better scalability. For sessions with hundreds of bundles, however, this strategy results in as many intermediate reconstructions and merging could become a bottleneck. To speed up this process, a separate merge job is initiated as soon as the first partial reconstruction is available and subsequent reconstructions are merged on-the- fly as they become available.

### Region-based constrained projection refinement

Unlike fully constrained refinement where all particles in a given tilt image have the same parameters, in our region-based approach particles are split into groups based on their position in 3D space. This allows modeling of beam-induced motion by estimating unique projection parameters for each individual region. The index functions $\mathcal{P}(i) : \{1 \cdots P\}$ and $\mathcal{G}(i) : \{1 \cdots G\}$ indicate the particle and region identities associated to image $I_i$, for a dataset with a total of $P$ particles and $G$ regions. The mapping $g$ between particle alignments in three dimensions and those corresponding to the 2D projections, can be written as:

$$[x_i, y_i, \theta_i, \phi_i, \psi_i]$$
$$= g\left(\alpha_{\mathcal{G}(i)}, \beta_{\mathcal{G}(i)}, x_{\mathcal{G}(i)}, y_{\mathcal{G}(i)}, \theta_{\mathcal{P}(i)}, \phi_{\mathcal{P}(i)}, \psi_{\mathcal{P}(i)}, x_{\mathcal{P}(i)}, y_{\mathcal{P}(i)}, z_{\mathcal{P}(i)}\right),$$

where $x_i, y_i, \theta_i, \phi_i, \psi_i$ are the translations and orientations assigned to particle projection $i$, $\alpha_{\mathcal{G}(i)}, \beta_{\mathcal{G}(i)}, x_{\mathcal{G}(i)}, y_{\mathcal{G}(i)}$ are the tilt-axis angle, tilt angle and image translations associated to region $\mathcal{G}(i)$ and $\theta_{\mathcal{P}(i)}, \phi_{\mathcal{P}(i)}, \psi_{\mathcal{P}(i)}, x_{\mathcal{P}(i)}, y_{\mathcal{P}(i)}, z_{\mathcal{P}(i)}$ represents the rigid transformation assigned to particle $\mathcal{P}(i)$. In region-based refinement, the overall projection matching objective function, $D$, is defined by accumulating cisTEM scores, $d$, calculated between images $I_i$ and reprojections of the model $R$, by optimizing over the geometric constraints $\alpha_g, \beta_g, x_g, y_g$:

$$D = \sum_{g=1}^{G} \sum_{i|\mathcal{G}(i)=g} d\left(I_i, R_{g(\alpha_g, \beta_g, x_g, y_g, \theta_{\mathcal{P}(i)}, \phi_{\mathcal{P}(i)}, \psi_{\mathcal{P}(i)}, x_{\mathcal{P}(i)}, y_{\mathcal{P}(i)}, z_{\mathcal{P}(i)})}\right),$$

where the second argument of the score function $d$ represents a reprojection of the model in the orientation given by the tilt geometry of each region and the orientation of particles contained in that region. The overall sum is arranged into $G$ groups (one per region), each having $P_g$ terms (one for each particle present in region $g$). For example, a tomogram can be divided into 4 × 4 × 2 regions (in $x$, $y$ and $z$) resulting in a total of $G = 32$ regions. Only particles located within a region are used for refining the micrograph parameters for that region. The number of regions can be adjusted to account for local motion at different spatial scales. Using more regions will result in finer spatial resolution, but the higher number of unknowns could potentially lead to overfitting. The use of fewer regions will result in lower spatial resolution for the sample deformation model, but has less chance of overfitting due to the reduced number of free parameters.

## Particle-based CTF refinement

Similar to region-based refinement, particle-based CTF refinement also requires defining a mapping $h$:

$$[df1_i, df2_i, Astig_i] = h\left(df1_{\mathcal{G}(i)}, df2_{\mathcal{G}(i)}, Astig_{\mathcal{G}(i)}\right),$$

between the defocus and astigmatism parameters assigned to individual 2D projections ($df1_i, df2_i, Astig_i$) and the mean defocus and astigmatism parameters associated to the region $\mathcal{G}$ ($i$): $df1_{\mathcal{G}(i)}, df2_{\mathcal{G}(i)}, Astig_{\mathcal{G}(i)}$. The projection matching objective function, $D$, is then maximized with respect to the CTF parameters of each region $df1_{\mathcal{G}(i)}, df2_{\mathcal{G}(i)}, Astig_{\mathcal{G}(i)}$:

$$D = \sum_{g=1}^{G} \sum_{i|\mathcal{G}(i)=g} d\left(I_i, R_{|h\left(df1_{\mathcal{G}(i)}, df2_{\mathcal{G}(i)}, Astig_{\mathcal{G}(i)}\right)}\right),$$

where the overall sum is arranged into groups (one per region), each having $P_g$ terms (one per particle located in region $g$). The maximization is performed using the Powell minimizer and the search is conducted within a user-specified tolerance for the defocus and astigmatism parameters.

## Video frame refinement and self-tuning exposure weighting

Frames for each particle projection are extracted from the raw videos of each tilt. Running averages for each frame are produced and used for alignment against the most current 3D reference. This process produces a set of noisy trajectories (one trajectory per tilted image, per particle), which we then regularize using spatial and temporal smoothness constraints to prevent overfitting[48]. To determine the weights of the exposure filter, score averages over all particles in each frame are calculated and used for exposure weighting using a modified version of cisTEM's reconstruct3d program[52]. Correlation scores assigned to individual frames obtained during processing of EMPIAR-10164 are shown in Fig. 3b. Corresponding 2D frequency weights were derived using the same formula used elsewhere[48], with the total number of frames corresponding to the number of frames per tilt, times the number of tilts (8 × 41 = 328; Fig. 3c). For tilt series that do not contain enough particles to produce reliable weights, a mean exposure curve can be obtained from the entire dataset and used for weighting (Fig. 3d). The very low doses per frame used in tomography can result in overfitting, usually manifested as FSC curves having a sharp falloff at the maximum resolution used for refinement[57]. To prevent this, longer running frame averages can be used to increase SNR and stronger spatial regularization of trajectories can be imposed on the movement of neighboring particles.

**Particle cleaning based on scores and spatial proximity.** One way to improve map resolution is to remove particles that do not contribute high-resolution information to the final reconstruction. These particles may correspond to damaged or false positives that were identified during particle picking, or to particles that cannot be properly aligned during refinement. To remove these from the downstream refinement, we calculated mean scores for each particle by averaging the scores from the tilted projections of each particle. A user-specified threshold was then used to remove particles that have scores lower than the cutoff. This produces cleaner particle sets and contributes to improving resolution. For large particles like ribosomes, a bimodal distribution of scores can be observed and in this case the threshold can be determined automatically as shown previously[58]. We also removed duplicate particles that were too close to their neighbors, using a user-specified minimum distance.

## Constrained classification of particle projections

Our strategy for heterogeneity analysis from 2D projections is based on the 3D single-particle classification algorithm implemented in cisTEM[52], with several changes to accommodate our distributed processing workflow and the imposition of the tilt geometry constraints. The randomization algorithm used to generate the $N$ discrete initial references was rewritten to account for the use of geometric constraints and to bundle images into groups representing unique sub-volumes. The local statistics of the refinement parameters (average and variance) resulting from our distributed workflow are different from the global statistics (obtained when considering all images in the dataset) and this can affect the classification results. To overcome this statistical bias, we always used the global refinement statistics when doing the distributed calculation of per-particle occupancies. The probabilities assigned during 3D reconstruction within each class also used the global refinement statistics. During the calculation of occupancies, Gaussian weights centered at the 0-degree tilt image with a s.d. of 6° were used to assign higher weights to the low tilt angle projections (the use of score-based weights in this case did not improve the classification results and instead led to overfitting due to the low-SNR conditions). All constrained classification runs were performed using the full range of tilts. The focused classification approach implemented in cisTEM can also be used within our constrained classification framework to determine structural flexibility restricted to a specific region or area of a complex.

## Validation of constrained classification using SP datasets

To validate our classification approach, we generated a test dataset by combining two single-particle datasets from HIV-1 Env samples with Fabs bound at different locations. Running averages of six video frames were considered as the equivalent of a tilt series with a similar dose but higher SNR because the images are not tilted. This synthetic mixture provides the necessary ground truth to test our classification algorithm. First, 3D classification was performed using video averages (without considering the frames) using standard single-particle classification routines implemented in cisTEM. The two classes were successfully separated after seven iterations. The misassignment errors in each class were 4.8% of particles for class 1 and 0.2% for class 2 (Extended Data Fig. 4a). Next, the dataset containing the same particle positions as before but now using the running frame averages for each particle was used for classification without imposing any constraints. The total number of projections in this case was ~60 times larger than the original dataset, which corresponded to the number of frames in the original micrographs. Even after 28 iterations, standard 3D classification as implemented in cisTEM failed to converge due to the lower SNR of the running frame averages. The percentages of misassigned particles for each class were 44.9% and 32.6%, respectively (Extended Data Fig. 4b). After applying the constraints to ensure that running frame averages belonging to the same particle were assigned to the same class, 3D classification was able to successfully separate the classes with only small assignment errors observed (3.0% and 3.8%, respectively; Extended Data Fig. 4c). The quality of the reconstructed maps was similar to the control results obtained by classifying the frame averages, demonstrating the effectiveness of our approach at classifying low-SNR particle projection sets.

## Book-keeping, metadata management and code optimization

The general philosophy behind nextPYP is to only keep copies of the raw data and the necessary metadata required to reproduce the final 3D reconstruction(s). This eliminates the need to store intermediate results (such as sub-volumes or particle stacks), which require considerable amounts of space and introduce I/O bottlenecks that slow down processing. This strategy results in a light-weight framework that can be scaled to analyze large datasets, while ensuring data reproducibility. Internally, metadata for tilt series is stored in binary pickle files and metadata generated during refinement is stored as extended .par files in compressed format using bzip2. To speed up processing, all routines for constrained refinement are implemented in C++ and built with the Intel compiler using MKL libraries. To avoid intermediate I/O steps and

speed up the evaluation of particle scores, our C++ code was linked against cisTEM's refine3d function[52].

## Compute resources and running times

nextPYP runs on Linux and uses containerization technology implemented in Apptainer (https://apptainer.org/) to ensure portability and reproducibility. Running the program in cluster mode requires access to a SLURM instance with a shared file system between the virtual machine running the web server and the compute nodes. Running times for pre-processing, high-resolution refinement and classification are reported in Supplementary Table 2 on a per-tilt series basis. If enough resources are available, all tilt series in a dataset can be run in parallel (bundle size of 1), making the total running time equal to the time per tilt series. If the bundle size is greater than 1, the running time will be the time per tilt series, times the bundle size. Running time during refinement (without considering reconstruction) is proportional to the number of tilts used for refinement. For fully constrained or region-based refinement, as well as 3D classification, memory consumption depends on the box size and the number of particles per tilt series. For video frame refinement, memory usage only depends on the box size. A minimum of 4 GB per vCPU is recommended for running most tasks.

## Ease-of-use, multi-user environment and interoperability

All results from data processing were streamed in real time to the GUI and could be conveniently accessed from multiple locations using a web-browser (Extended Data Fig. 9). To account for different computer setups and data processing needs, nextPYP can operate in standalone mode or in cluster mode. In standalone mode, the web server and all processing jobs run locally on the same virtual machine, whereas in cluster mode, all processing jobs are submitted to an HPC cluster environment providing scale out capabilities. This mode is especially useful for supporting on-the-fly data pre-processing sessions acquired using BISECT where multiple tilt series are acquired in parallel[13] and to refine structures from datasets with thousands of tilt series. For training its deep-learning models for particle picking, nextPYP needs access to a GPU. All other operations are currently executed on CPU cores using multiple threads (GPU support will be expanded in the future). The program also allows importing user-defined sequences of processing blocks, called 'workflows' that can be used to automate routine data analysis tasks or run standardized benchmarks. Multiple users and groups are supported by providing project-specific access control and allowing users to share sessions or projects within a group or with other users. When system-wide data access restrictions are required: (1) Linux groups and permissions can be used to limit access to certain users, or (2) multiple instances of nextPYP can be executed (one for each user or group). In addition to the GUI, a command line interface is also provided that allows finer control of all data processing commands. To facilitate interoperability with other packages, nextPYP has functions to import and export metadata in the .star format used by recent versions of cisTEM[52], Relion[23] and M[37].

## Support for single-particle data processing

nextPYP also supports online and offline processing of cryo-EM micrographs. For online processing, raw video frames can be transferred from the microscope and analyzed on-the-fly using a similar user interface as that used for tilt series. In addition to video frame alignment, CTF determination and particle picking (size-based and neural network-based approaches), nextPYP can also run 2D classification using wrappers around cisTEM[52]. Metadata resulting from online sessions can be exported in .star format to external packages for further processing. Users can also apply filters and export subsets of micrographs according to user-defined criteria such as CTF-fit, maximum drift and number of particles. For offline processing,

nextPYP supports reference-based 3D refinement and 3D classification jobs[52], as well as reference-based video frame refinement and data-driven exposure weighting[48]. All single-particle operations use the same scalable and storage-efficient approach used for SP-CET, where particle and frame stacks are only saved temporarily in local scratch. A fully featured GUI provides interactive visualization of micrographs, CTF, maps and refinement statistics. Users can also pick particles interactively to train and apply semi-supervised neural network models[44]. Micrographs, videos and metadata can be imported/exported in .star format to use with external packages. Shape masking and map post-processing operations are also supported.

## Reporting summary

Further information on research design is available in the Nature Portfolio Reporting Summary linked to this article.

## Data availability

This study utilized raw tilt series available from the EMPIAR database under accession nos. 10064, 10164, 10499, 10987 and 11273 and cryo-EM maps available from the Electron Microscopy Data Bank (EMDB) under accession nos. 8803, 11638, 11650, 11655, 16209 and 33118. Cryo-EM density maps produced in this study were deposited in the EMDB under accession nos. EMD-41196 and EMD-41197 for EMPIAR-10164 (five tilt series and full dataset), EMD-41199 for EMPIAR-11273, EMD-41205, EMD-41207, EMD-41210, EMD-41211 and EMD-41212 for EMPIAR-10064 (classes 1, 2, 3, 4 and 5), EMD-41220, EMD-41221 and EMD-41222 for EMPIAR-10499 (classes 1, 2 and 3) and EMD-41223, EMD-41224, EMD-41225, EMD-41226 and EMD-41227 for EMPIAR-10987 (classes 1, 2, 3, 4 and 5).

## Code availability

The open-source code for nextPYP is available at https://github.com/nextpyp.

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

## Acknowledgements

Research reported in this publication was supported by the National Institute of General Medical Sciences of the National Institutes of Health under award no. R01GM141223 to A.B. The content is solely the responsibility of the authors and does not necessarily represent the views of the National Institutes of Health. This study has been made possible in part by CZI grant DAF2021-234602 and grant https://doi.org/10.37921/181830egjglp from the Chan Zuckerberg Initiative DAF, an advised fund of Silicon Valley Community Foundation (funder DOI 10.13039/100014989). This study was partially supported by the National Institute of Allergy and Infectious Diseases of the National Institutes of Health under award no. U54-AI170752 to A.B. This study utilized the computational resources offered by Duke Research Computing (http://rc.duke.edu). We thank T. Futhey, C. Kneifel,

K. Kilroy, M. Newton, V. Orlikowski, T. Milledge, J. Dorff, Z. Hill and D. Lane from the Duke Office of Information Technology and Research Computing for providing assistance with the computing environment. We thank J. Bouvette, K. Sharma, M. Borgnia, L. He and A. Watson for providing feedback on an early prototype of nextPYP. We thank P. Acharya and R. Henderson for providing access to the single-particle datasets of HIV-1 Env used to validate our constrained classification approach.

## Author contributions

H.L., Y.Z., W.J., Q.H., X.D. and A.B. developed and implemented the command line tool. J.M., J.P., J.M. and A.B. developed and implemented nextPYP. H.L., Y.Z. and A.B. carried out data processing. H.L., Y.Z. and A.B. contributed to writing the paper and prepared the figures. A.B. conceived the project.

## Competing interests

The authors declare no competing interests.

## Additional information

**Extended data** is available for this paper at https://doi.org/10.1038/s41592-023-02045-0.

**Correspondence and requests for materials** should be addressed to Alberto Bartesaghi.

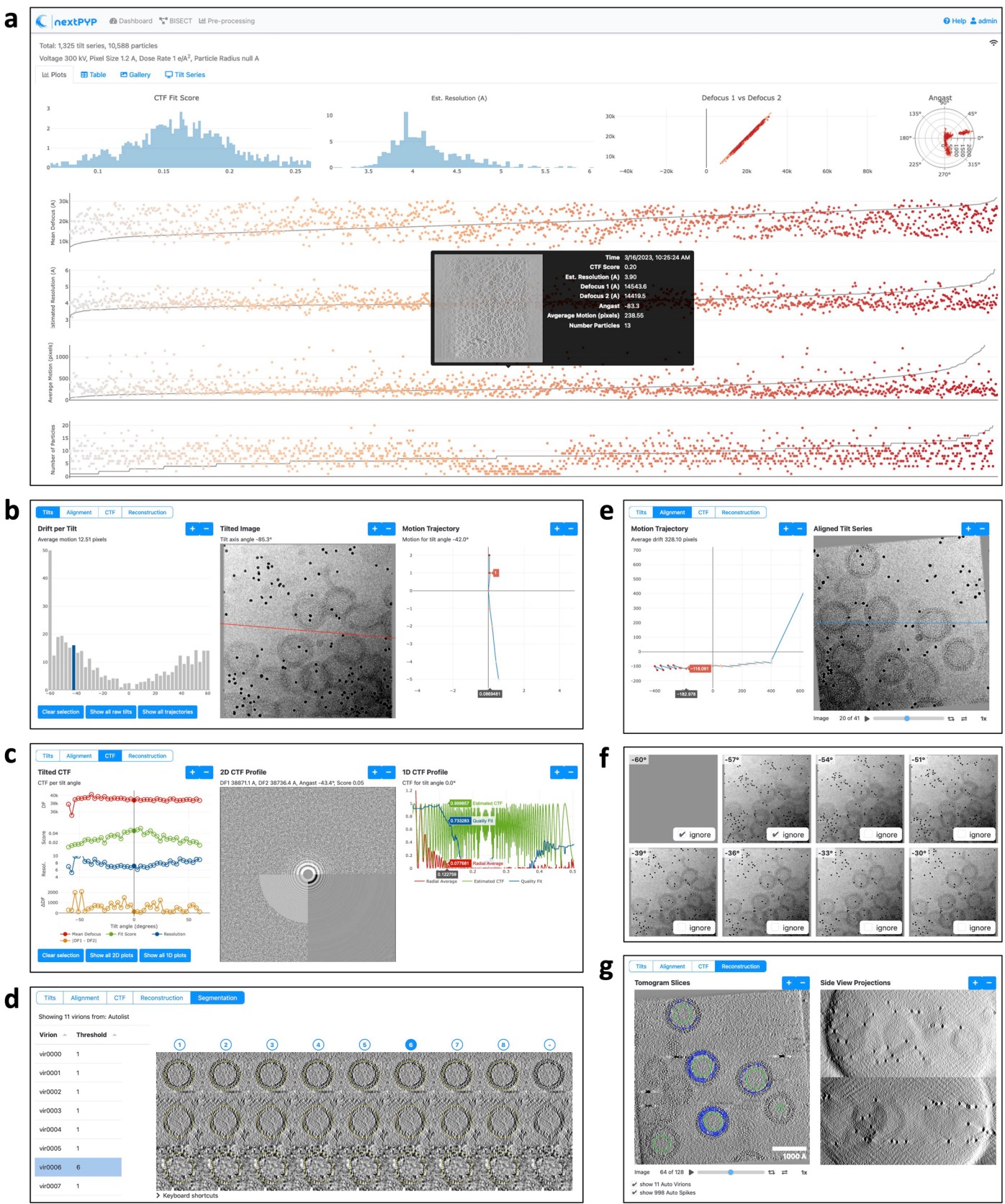

**Extended Data Fig. 1 | nextPYP graphical user interface (GUI) for on-the-fly monitoring of tilt series.** nextPYP provides visualization tools to monitor data quality during data acquisition. Results are streamed in real-time to a web-based interface providing convenient access to various metrics of image and reconstruction quality. The GUI includes components to visualize overall dataset statistics (**a**), video frame alignment (**b**), per-tilt CTF estimation (**c**), virion segmentation (**d**), tilt series alignment (**e**), gallery view to discard individual tilt-images (**f**), and 3D particle picking (**g**) (virion and particle positions shown in the left panel were calculated for all 43 tilt series from EMPIAR-10164).

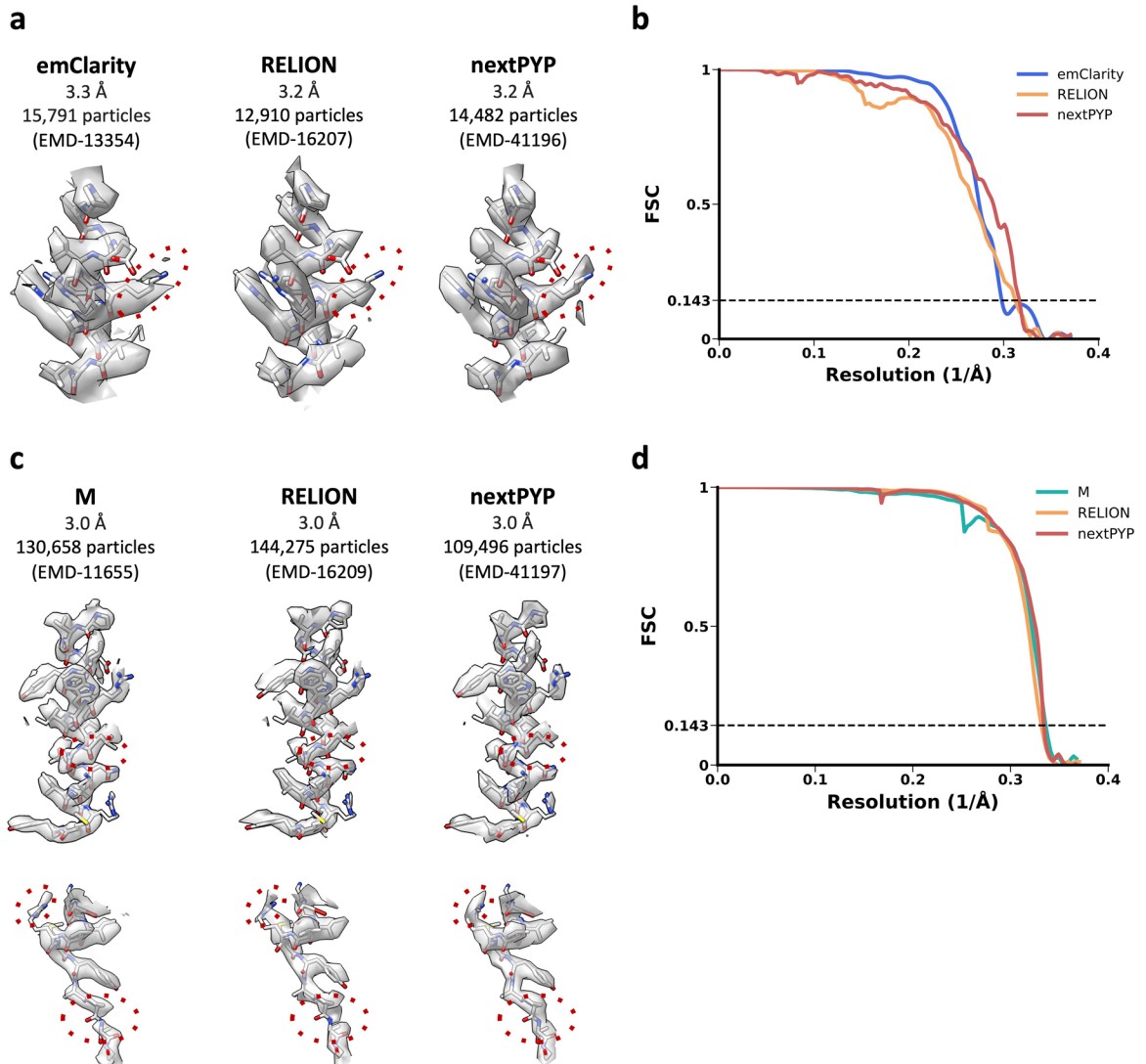

**Extended Data Fig. 2 | Comparison of final map resolution against other packages (EMPIAR-10164). a**) Alpha helix segment with fitted atomic coordinates PDB ID 5l93 showing maps obtained from a subset of 5 tilt series using emClarity (3.3 Å, 15,791 particles), RELION (3.2 Å, 12,910 particles) and nextPYP (3.2 Å, 14,482 particles). Improved map features are highlighted in red. **b**) Corresponding FSC curves between half-maps showing estimated resolutions using the 0.143 cutoff criteria. **c**) Alpha helix segments with fitted atomic coordinates for maps obtained from the entire dataset (43 tilt series) using M (3.0 Å, 130,658 particles), RELION (3.0 Å, 144,275 particles), and nextPYP (3.0 Å, 109,496 particles). Improved map features and density continuity are highlighted in red. d) Corresponding FSC curves between half-maps.

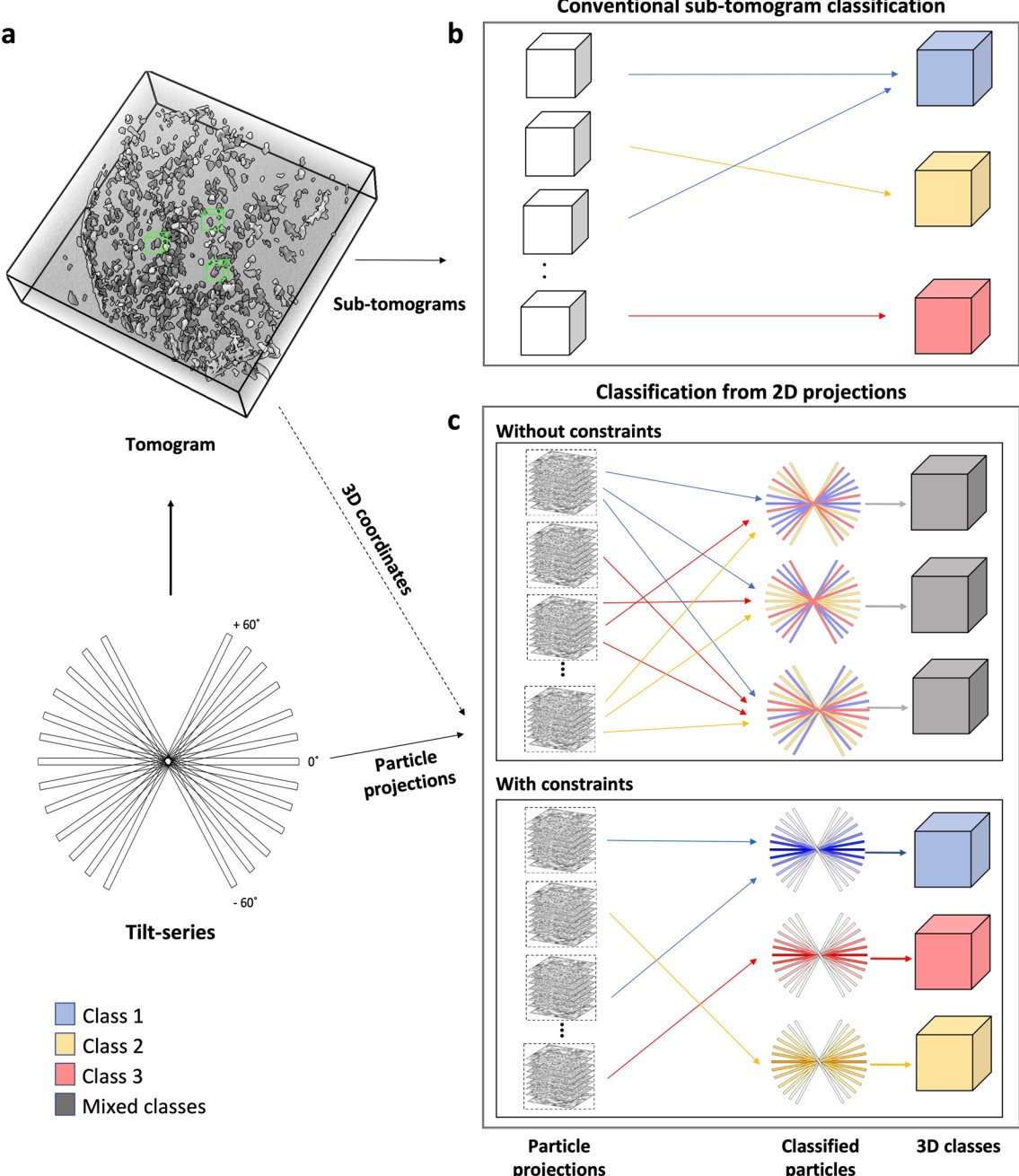

**Extended Data Fig. 3 | Constrained classification of tilted particle projections.** Strategy for 3D classification of single-particle cryo-electron tomography data. **a**) In conventional sub-tomogram classification, tomograms are first reconstructed and sub-volumes are extracted for subsequent 3D classification. Sub-volumes are sorted into 3 classes, corresponding to different conformational states (blue, yellow, and red). **b**) Classification from

2D projections requires extraction of 2D images from the raw tilt series data using the coordinates of particles in 3D. Classification without imposing the constraints of the tilt geometry incorrectly classifies the particles due to the low SNR of tomographic particle projections. **c**) Imposition of the constraints allows correct separation of multiple conformations.

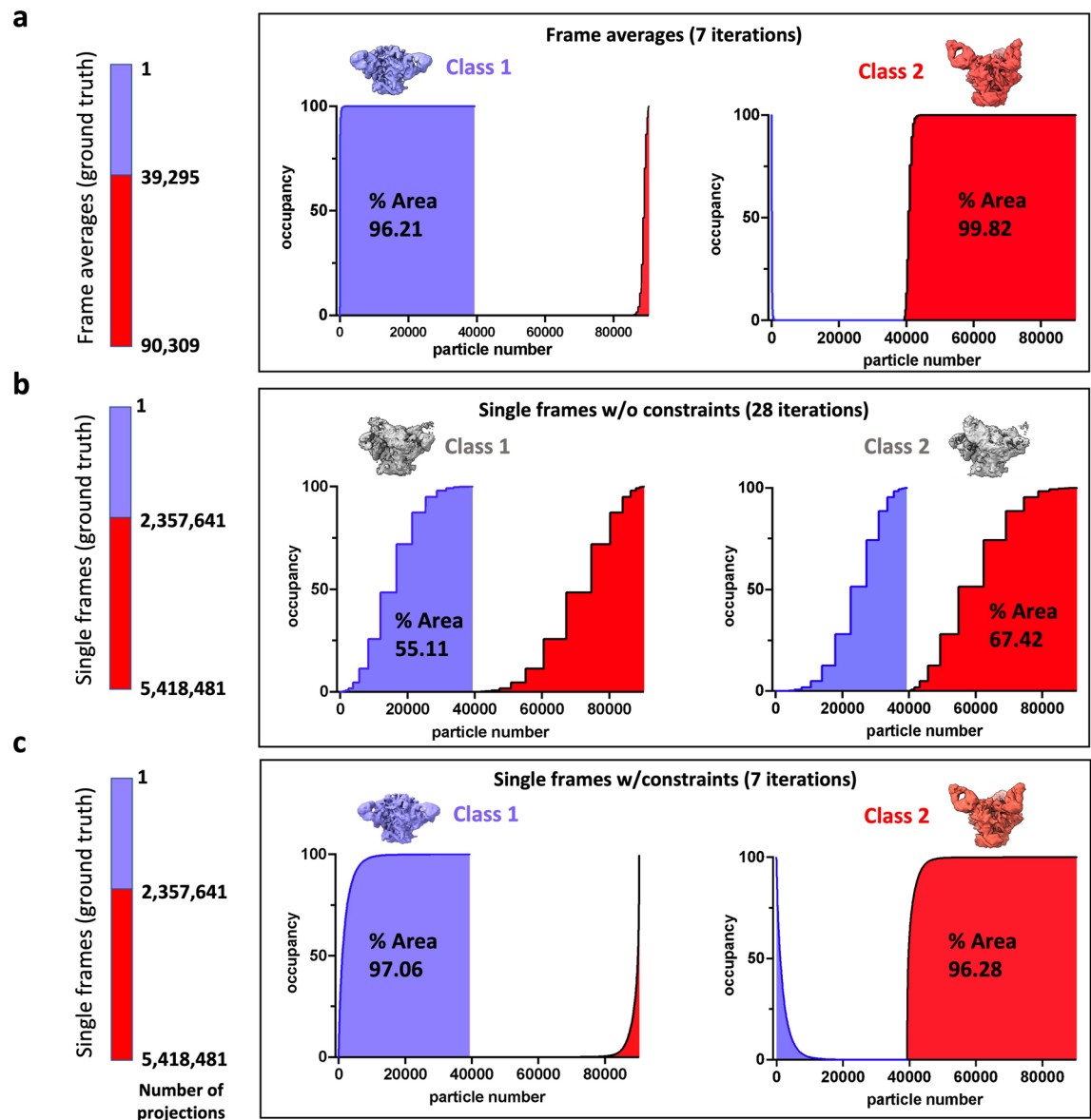

**Extended Data Fig. 4 | Validation of constrained classification using a synthetic mixture of single-particle datasets.** Micrographs from two datasets of HIV-1 Env bound to different antibodies were combined and subjected to 3D classification. **a**) Baseline classification using video frame averages. Particle ground-truth distribution is shown with a colored bar on the left (Class 1 blue, Class 2 red). Area plots on the right show the occupancy distribution as a function of the particle number for each class. Reconstructed maps for each class after 7 iterations are shown above the plots showing correct separation of the two conformations. **b**) Classification using video frames without imposing constraints after 28 iterations fails to separate the two states. **c**) Constrained classification using video frames correctly recovers the two conformations after 7 iterations.

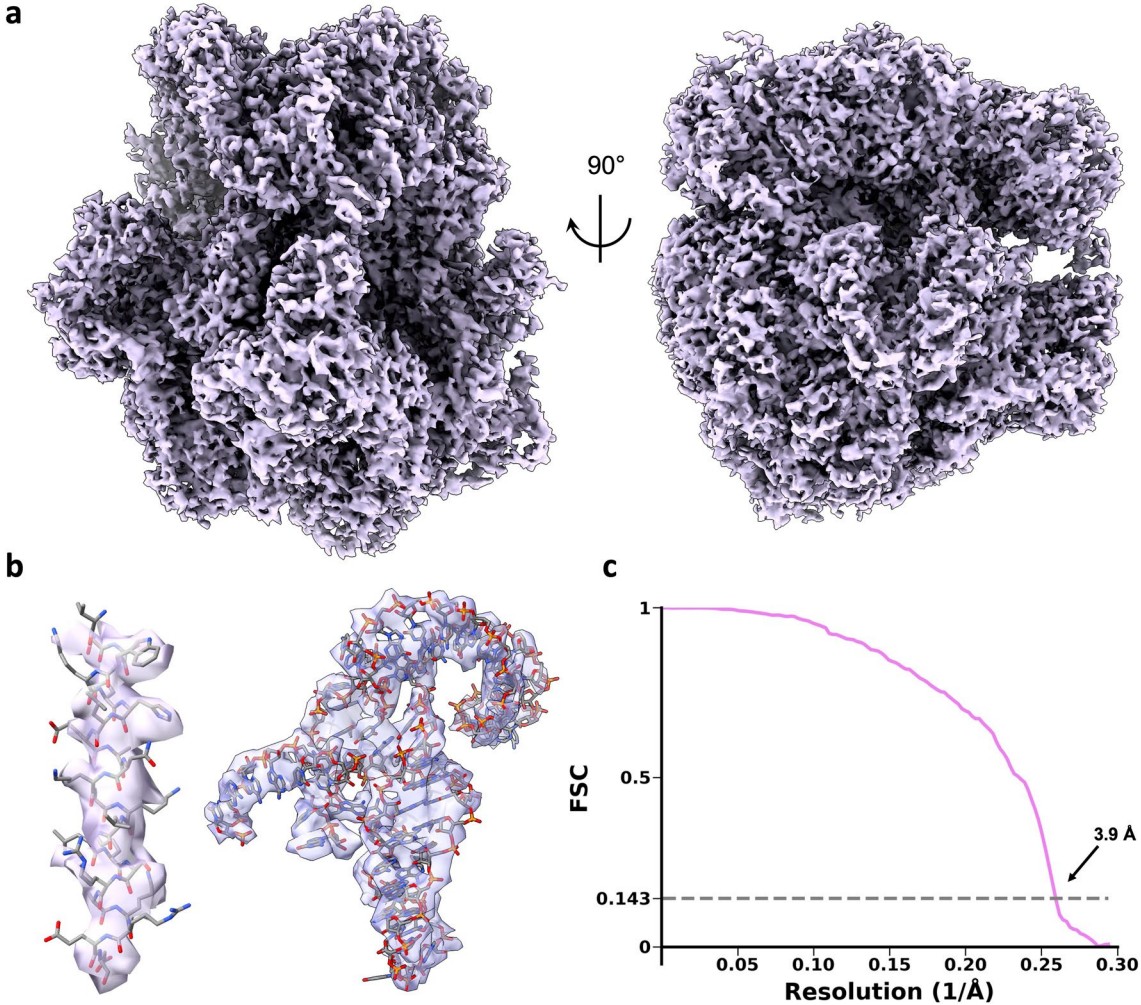

**Extended Data Fig. 5 | Consensus map obtained from tilt series of in-cell 70S ribosomes (EMPIAR-10499). a**) Overview of 3.9 Å resolution reconstruction obtained from 18,135 particles (without performing particle cleaning since this dataset was used to benchmark the constrained classification strategy). **b**) Alpha helical and RNA helical segments shown with model fit into map. **c**) FSC plot between half-maps (0.143 cutoff criteria).

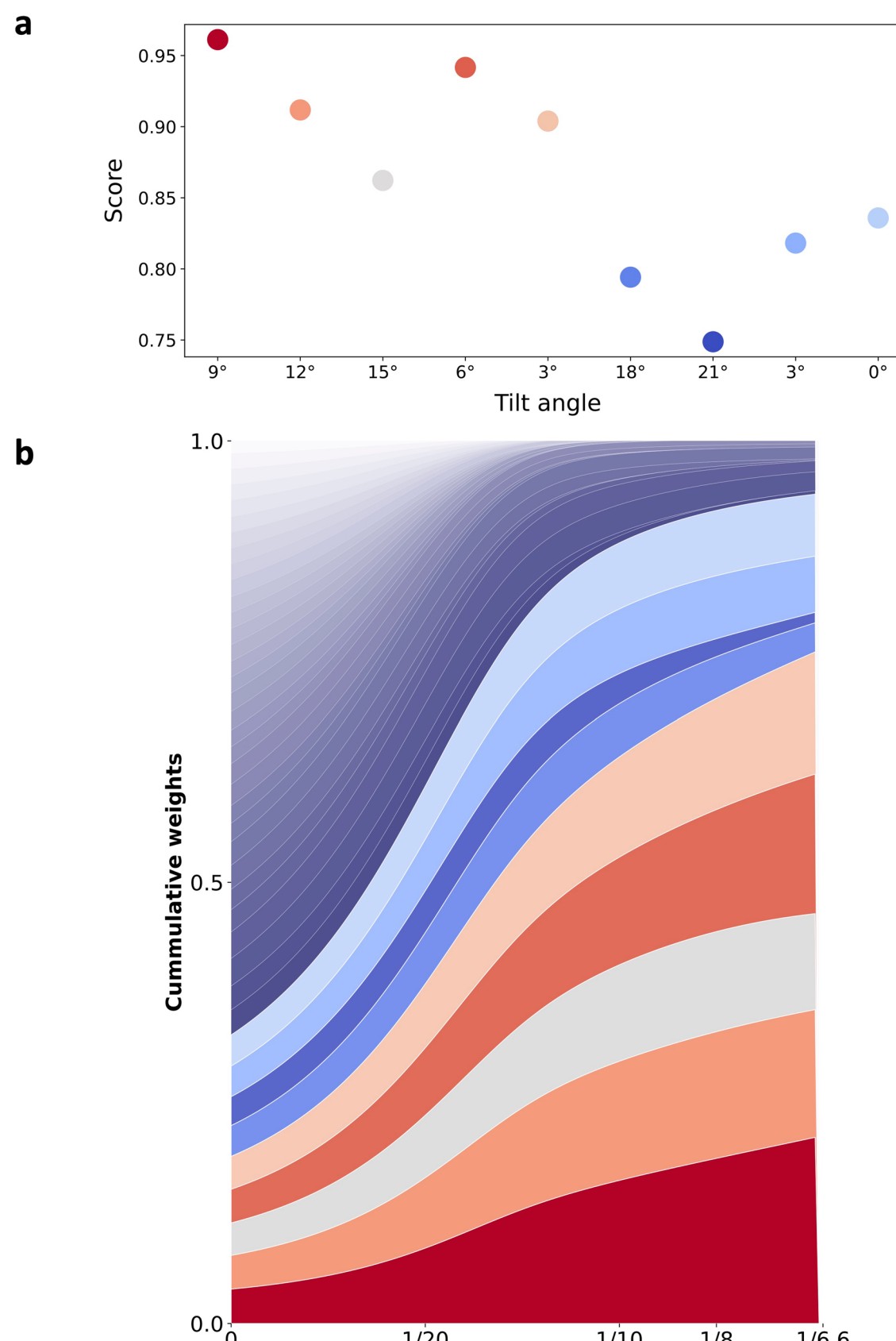

**Extended Data Fig. 6 | Per-tilt frequency dependent weights for 80S ribosomes from cryo-FIB lamellae (EMPIAR-10987). a)** Data-driven average similarity scores measured between the individual tilts of each particle and the 3D reference. **b)** Corresponding weights for individual tilts are shown as a function of frequency. Colors represent the relative contribution of each tilt from highest (dark red) to lowest (deep blue). Bands for each tilt are ordered in the sequence they were acquired (bottom=first, top=last).

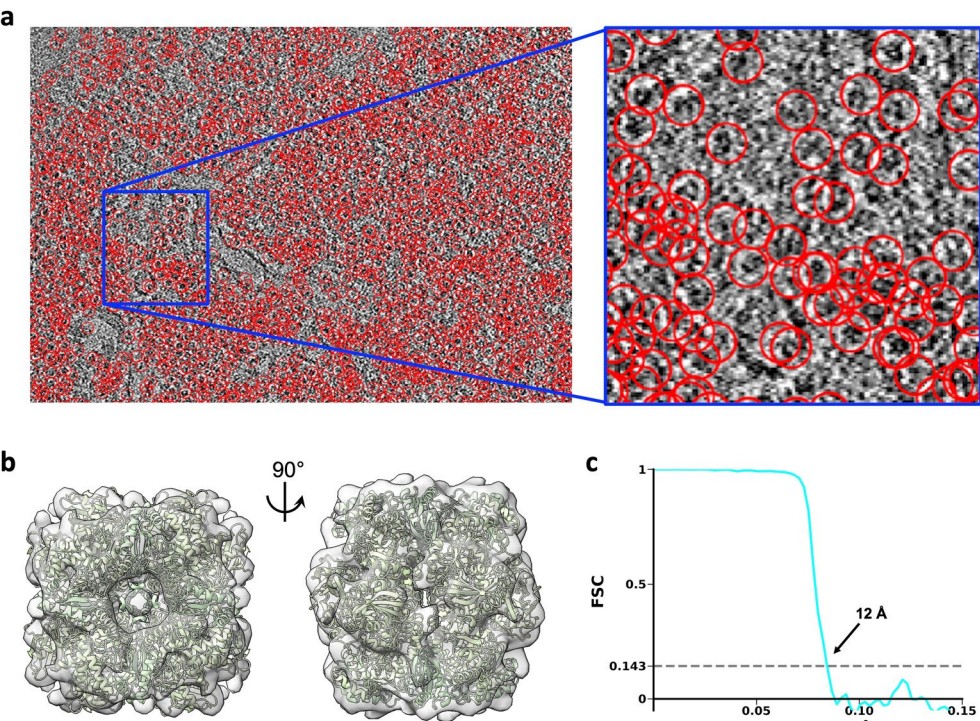

**Extended Data Fig. 7 | Neural network-based particle picking of RuBisCO particles from tomograms of Chlamydomonas reinhardtii cells (EMPIAR-10694). a)** Slice through tomogram showing cellular features and selected RuBisCo particles (inset). **b)** 3D reconstruction obtained from 35,352 particles with atomic model fit into map. **c)** FSC plot between half-maps showing a resolution of 12 Å (0.143 cutoff criteria).

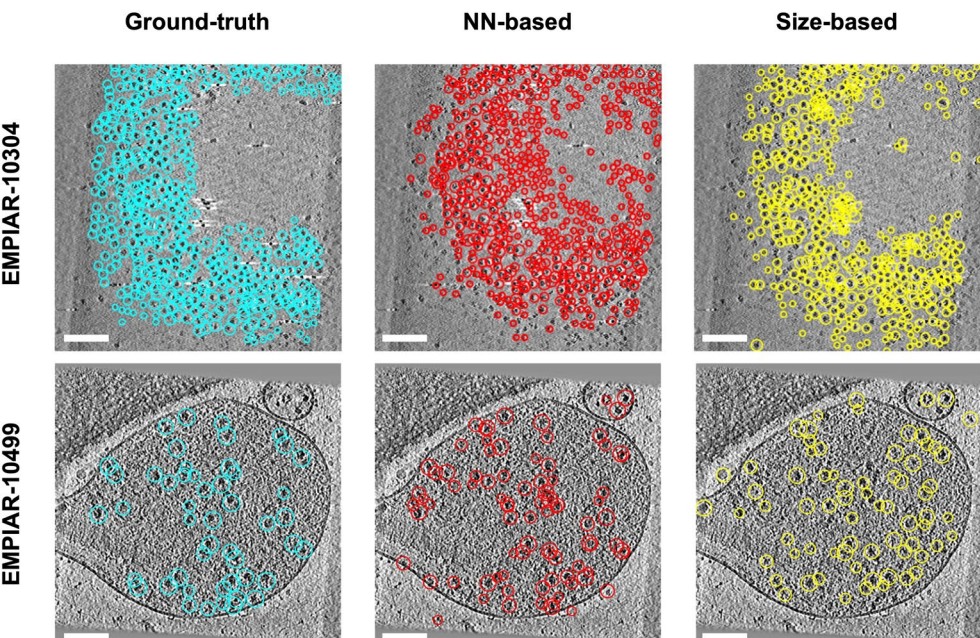

**Extended Data Fig. 8 | Comparison between size-based and neural network-based particle picking approaches.** Particle picking results on tomograms of ribosomes from EMPIAR-10304 (in vitro) and EMPIAR-10499 (in situ). Ground-truth particle positions (left, cyan) were obtained by manual picking. Corresponding results of neural network-based picking (middle, red) and size-based picking (right, yellow) are shown. Similar results were obtained for each of the 12 tilt series from EMPIAR-10304 and the 65 tilt series from EMPIAR-10499. Additional accuracy statistics are presented in Supplementary Table 5. Scale bars for EMPIAR-10304 are 200 nm and for EMPIAR-10499 are 100 nm.

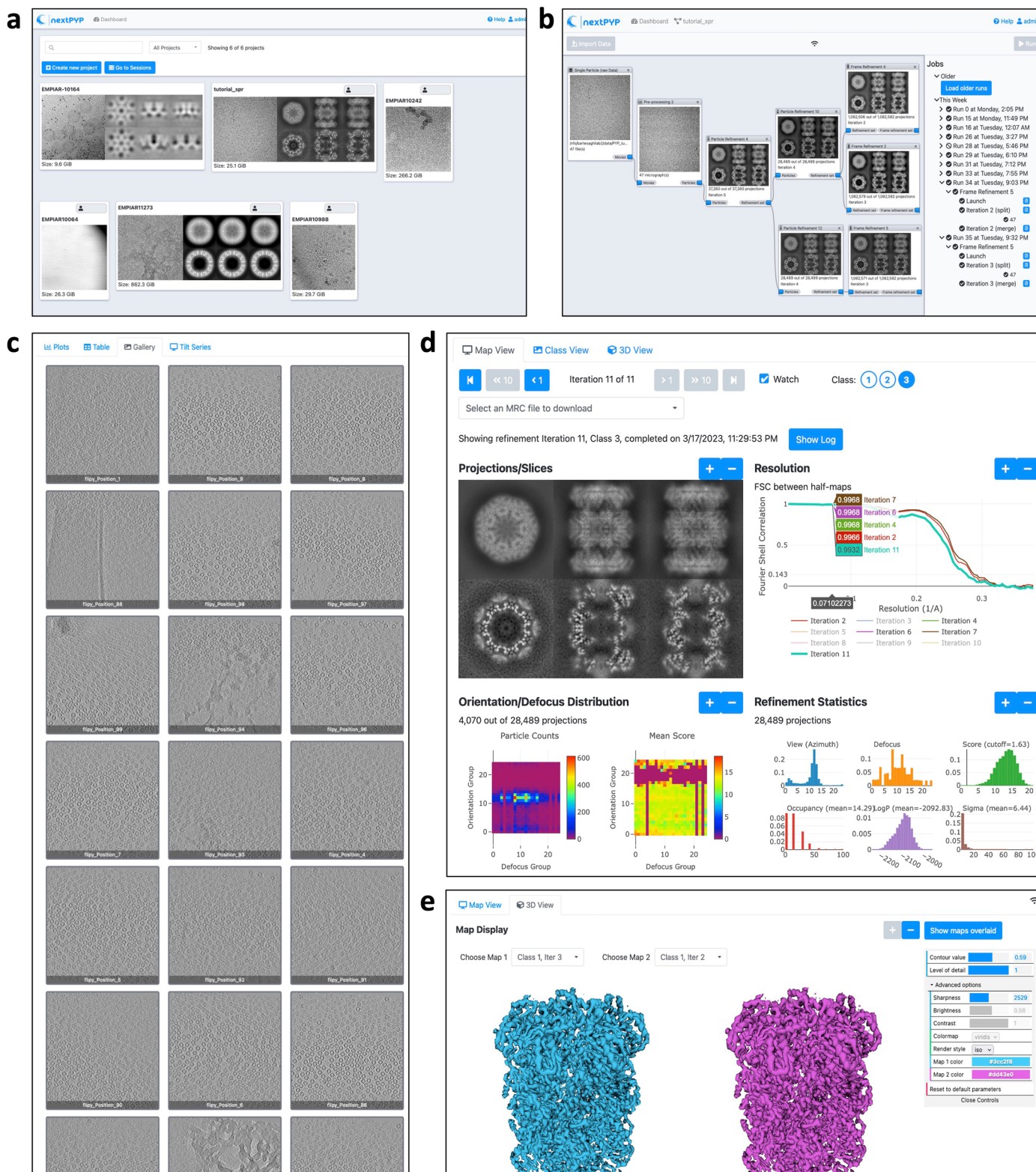

**Extended Data Fig. 9 | GUI components in nextPYP for high-resolution refinement and classification. a**) The main application dashboard shows information about multiple projects. **b**) Each project consists of multiple data processing blocks and access to log files and processing history is shown on the right. **c**) Gallery view shows multiple tomogram slices to facilitate inspection of large datasets. **d**) The refinement view shows metrics for reconstruction quality including map slices and projections, FSC plots and orientation distribution of particles for each iteration. Navigation controls provide access to maps from different iterations and classes. **e**) Multiple reconstructions can be displayed interactively in 3D.

# Reporting Summary

## Statistics

For all statistical analyses, confirm that the following items are present in the figure legend, table legend, main text, or Methods section.

| n/a | Confirmed | |
|---|---|---|
| ☐ | ☒ | The exact sample size (*n*) for each experimental group/condition, given as a discrete number and unit of measurement |
| ☐ | ☒ | A statement on whether measurements were taken from distinct samples or whether the same sample was measured repeatedly |
| ☒ | ☐ | The statistical test(s) used AND whether they are one- or two-sided *Only common tests should be described solely by name; describe more complex techniques in the Methods section.* |
| ☒ | ☐ | A description of all covariates tested |
| ☒ | ☐ | A description of any assumptions or corrections, such as tests of normality and adjustment for multiple comparisons |
| ☒ | ☐ | A full description of the statistical parameters including central tendency (e.g. means) or other basic estimates (e.g. regression coefficient) AND variation (e.g. standard deviation) or associated estimates of uncertainty (e.g. confidence intervals) |
| ☒ | ☐ | For null hypothesis testing, the test statistic (e.g. *F*, *t*, *r*) with confidence intervals, effect sizes, degrees of freedom and *P* value noted *Give P values as exact values whenever suitable.* |
| ☒ | ☐ | For Bayesian analysis, information on the choice of priors and Markov chain Monte Carlo settings |
| ☐ | ☒ | For hierarchical and complex designs, identification of the appropriate level for tests and full reporting of outcomes |
| ☒ | ☐ | Estimates of effect sizes (e.g. Cohen's *d*, Pearson's *r*), indicating how they were calculated |

*Our web collection on statistics for biologists contains articles on many of the points above.*

## Software and code

Policy information about availability of computer code

| Data collection | No new data collection was performed for this study. |
|---|---|
| Data analysis | Publicly available software used for data analysis: IMOD v4.11.12, cisTEM v2.0.0-alpha, CTFFIND v4.0.17, ITK v4.2.1, and VTK v5.10.1. Custom software developed for this study: nextPYP v0.5.0 (http://nextpyp.app). |

For manuscripts utilizing custom algorithms or software that are central to the research but not yet described in published literature, software must be made available to editors and reviewers. We strongly encourage code deposition in a community repository (e.g. GitHub). See the Nature Portfolio guidelines for submitting code & software for further information.

## Data

Policy information about availability of data

All manuscripts must include a data availability statement. This statement should provide the following information, where applicable:
- Accession codes, unique identifiers, or web links for publicly available datasets
- A description of any restrictions on data availability
- For clinical datasets or third party data, please ensure that the statement adheres to our policy

This study utilized raw tilt-series available from the Electron Microscopy Public Image Archive (EMPIAR) database under accession numbers 10064, 10164, 10499, 10987 and 11273 and cryo-EM maps available from the Electron Microscopy Data Bank (EMDB) under accession numbers 8803, 11638, 11650, 11655, 16209, and 33118. Cryo-EM density maps were deposited in the EMDB under accession numbers EMD-41196 and EMD-41197 for EMPIAR-10164 (5 tilt-series and entire

dataset), EMD-41199 for EMPIAR-11273, EMD-41205, EMD-41207, EMD-41210, EMD-41211, and EMD-41212 for EMPIAR-10064 (classes 1, 2, 3, 4 and 5), EMD-41220, EMD-41221, and EMD-41222 for EMPIAR-10499 (classes 1, 2 and 3), and EMD-41223, EMD-41224, EMD-41225, EMD-41226, and EMD-41227 for EMPIAR-10987 (classes 1, 2, 3, 4 and 5).

## Human research participants

Policy information about studies involving human research participants and Sex and Gender in Research.

| | |
|---|---|
| Reporting on sex and gender | No Human research participants were used in this study. |
| Population characteristics | N/A |
| Recruitment | N/A |
| Ethics oversight | N/A |

Note that full information on the approval of the study protocol must also be provided in the manuscript.

# Field-specific reporting

Please select the one below that is the best fit for your research. If you are not sure, read the appropriate sections before making your selection.

☒ Life sciences          ☐ Behavioural & social sciences          ☐ Ecological, evolutionary & environmental sciences

For a reference copy of the document with all sections, see nature.com/documents/nr-reporting-summary-flat.pdf

# Life sciences study design

All studies must disclose on these points even when the disclosure is negative.

| | |
|---|---|
| Sample size | All raw data used in this study were obtained from datasets deposited in the EMPIAR database. The total number of tilt-series available for each entry was used, as follows: EMPIAR-10064 (4 tilt-series), EMPIAR-10164 (43 tilt-series),EMPIAR-10304 (12 tilt-series), EMPIAR-10499 (65 tilt-series), EMPIAR-10694 (1 tilt-series), 10987 (20 tilt-series) and EMPIAR-11273 (100 tilt-series). |
| Data exclusions | All data from the corresponding entries were used for processing. No data were excluded from the analyses. |
| Replication | Detailed protocols for obtaining the final maps were documented and used by at least three different lab members to reproduce the results. Replication attempts resulted in maps with identical resolutions. |
| Randomization | Randomization was not relevant in this study. All tilt-series were used for the analyses so no randomization was needed. |
| Blinding | Blinding was not relevant for this study because all tilt-series were used for the analyses. No data were left out. |

# Reporting for specific materials, systems and methods

We require information from authors about some types of materials, experimental systems and methods used in many studies. Here, indicate whether each material, system or method listed is relevant to your study. If you are not sure if a list item applies to your research, read the appropriate section before selecting a response.

### Materials & experimental systems

| n/a | Involved in the study |
|---|---|
| ☒ | ☐ Antibodies |
| ☒ | ☐ Eukaryotic cell lines |
| ☒ | ☐ Palaeontology and archaeology |
| ☒ | ☐ Animals and other organisms |
| ☒ | ☐ Clinical data |
| ☒ | ☐ Dual use research of concern |

### Methods

| n/a | Involved in the study |
|---|---|
| ☒ | ☐ ChIP-seq |
| ☒ | ☐ Flow cytometry |
| ☒ | ☐ MRI-based neuroimaging |

