## [Peer Review File · Nature Methods]

Peer Review Information

Manuscript Title: nextPYP: a comprehensive and scalable platform for characterizing protein variability in-situ using single-particle cryo-electron tomography

Corresponding author name(s): Alberto Bartesaghi

Editorial Notes: n/a

Reviewer Comments & Decisions:

Decision Letter, initial version:

Dear Alberto,

Your Article, "nextPYP: a comprehensive and scalable platform for characterizing protein variability in-situ using single-particle cryo-electron tomography", has now been seen by three reviewers. As you will see from their comments below, although the reviewers find your work of considerable potential interest, they have raised a number of concerns. We are interested in the possibility of publishing your paper in Nature Methods, but would like to consider your response to these concerns before we reach a final decision on publication.

We therefore invite you to revise your manuscript to address these concerns. We ask you to focus on clarifying the technical questions raised by the reviewers regarding the workflow, doing a bit more to demonstrate the particle picking (refs 1 and 3), addressing or at least discussing possible data security issues (ref 3) that could come from the implementation, and clarifying which steps of the workflow are new to this work.

* include a point-by-point response to the reviewers and to any editorial suggestions

* please underline/highlight any additions to the text or areas with other significant changes to facilitate review of the revised manuscript

* address the points listed described below to conform to our open science requirements

* ensure it complies with our general format requirements as set out in our guide to authors at www.nature.com/naturemethods

* resubmit all the necessary files electronically by using the link below to access your home page

[Redacted] This URL links to your confidential home page and associated information about manuscripts you may have submitted, or that you are reviewing for us. If you wish to forward this email to co-authors, please delete the link to your homepage.

We hope to receive your revised paper within two months. If you cannot send it within this time, please let us know. In this event, we will still be happy to reconsider your paper at a later date so long as nothing similar has been accepted for publication at Nature Methods or published elsewhere.

OPEN SCIENCE REQUIREMENTS

REPORTING SUMMARY AND EDITORIAL POLICY CHECKLISTS

Please note that these forms are dynamic ‘smart pdfs’ and must therefore be downloaded and completed in Adobe Reader. We will then flatten them for ease of use by the reviewers. If you would like to reference the guidance text as you complete the template, please access these flattened versions at <http://www.nature.com/authors/policies/availability.html>.

DATA AVAILABILITY

All novel DNA and RNA sequencing data, protein sequences, genetic polymorphisms, linked genotype and phenotype data, gene expression data, macromolecular structures, and proteomics data must be deposited in a publicly accessible database, and accession codes and associated hyperlinks must be provided in the “Data Availability” section.

Please include a “Data availability” subsection in the Online Methods. This section should inform readers about the availability of the data used to support the conclusions of your study, including accession codes to public repositories, references to source data that may be published alongside the paper, unique identifiers such as URLs to data repository entries, or data set DOIs, and any other statement about data availability. At a minimum, you should include the following statement: “The data that support the findings of this study are available from the corresponding author upon request”, describing which data is available upon request and mentioning any restrictions on availability. If DOIs are provided, please include these in the Reference list (authors, title, publisher (repository name),

identifier, year). For more guidance on how to write this section please see:

<http://www.nature.com/authors/policies/data/data-availability-statements-data-citations.pdf>

CODE AVAILABILITY

Please include a “Code Availability” subsection in the Online Methods which details how your custom code is made available. Only in rare cases (where code is not central to the main conclusions of the paper) is the statement “available upon request” allowed (and reasons should be specified).

ORCID

Nature Methods is committed to improving transparency in authorship. As part of our efforts in this direction, we are now requesting that all authors identified as ‘corresponding author’ on published papers create and link their Open Researcher and Contributor Identifier (ORCID) with their account on the Manuscript Tracking System (MTS), prior to acceptance. This applies to primary research papers only. ORCID helps the scientific community achieve unambiguous attribution of all scholarly contributions. You can create and link your ORCID from the home page of the MTS by clicking on ‘Modify my Springer Nature account’. For more information please visit www.springernature.com/orcid.

Sincerely,
Rita

Rita Strack, Ph.D.

Senior Editor
Nature Methods

Reviewers' Comments:

Reviewer #1:

Remarks to the Author:

Liu and colleagues present a new end-to-end framework for processing, reconstruction, and 3D analysis of electron cryo-tomography data. The software has a user-friendly GUI-based interface and provides multiple interactive ways to visualize data, processing workflows, and results. Importantly, the package is all-inclusive and will give researchers a quick low-barrier entry option into the field of subtomogram analysis. Overall, this is an important contribution to the electron cryo-tomography data analysis ecosystem which at present lacks a unified processing solution and has relied on custom workflows using a mélange of software tools. The manuscript is well-written and gives mostly satisfactory descriptions of the algorithms and approaches used in the system. The work will be of particular interest to researchers who are already using cryo-tomography structural analysis or have been waiting for the technique to become more accessible before investing time into it. The prospects of improving the convenience and throughput of cryo-tomography will also intrigue the broader structural biology, disease studies, and drug development audiences. Therefore, I believe that the paper is a good match for Nature Methods and pending a few comments and questions, I enthusiastically recommend it for publication.

1. The version of nextPYP described in the paper uses mainly CPU computation. GPUs have proven to be significantly more efficient in terms of cost and overall speed for cryo-EM data analysis. With increasing dataset sizes and limited access to CPU-based clusters at many research institutions, GPUs may be the only processing option for many labs. Please add a comment in the paper about the possibilities or limitations of transitioning nextPYP to GPU compute.
2. Related to the above comment, please add a row in the Platform section of Supplementary Table 1 about GPU compute support.
3. The region-based constrained geometry refinement relies on a good number and distribution of particles in the sample. This is not always the case in situ. How do you determine the number/size of regions? Can they be automatically and/or non-uniformly distributed based on the local particle density? Did you test different region sized on any of your datasets? Is it possible to use more abundant particles like ribosomes for the region-based refinement and use these parameters for the particle alignment of other species similar to M? Please comment on these points in the paper.

4. Does nextPYP support “hands-off” processing (on-the-fly or post-acquisition) by setting the necessary parameters initially and then letting it work through the data? If not, would it be possible in the future to develop nextPYP-based workflows that do not require interactive steps?
5. Why were the global reconstructions of the EMPIAR-10064 and EMPIAR-10499 not included in the results? In particular, a global reconstruction of EMPIAR-10499 comparing the performance of nextPYP and M would be very interesting.
6. The reconstructions of three of the six datasets presented used a very narrow tilt range (-6 to +6 deg, Supplementary Table 3). Please add an explanation about that, the selection process of these tilt ranges, and the impact of the size of the processed tilt range on the processing time.
7. The particle cleaning uses a user-specified particle score cutoff. How is the optimal value selected? (visually?) Are there possible ways to determine the optimal cutoff value automatically?
8. The constrained particle projection classification applies an empirical Gaussian upweighting of low tilt angle images which seems rather arbitrary. Can the image weighting be based on the frame scores from the 3D reconstruction (Fig. 3C)?
9. The “ab-initio” pose estimation approach is in fact reference-based. This is a bit confusing. What references were used and to what extent were they low-pass filtered?
10. It is stated that the size-based particle picking approach can facilitate particle picking of large complexes like ribosomes in-situ, yet the ribosome datasets shown here were picked with the deep learning approach. It would be interesting to see a comparison between the two approaches for in vitro and in situ data.
11. For the virion segmentation. What parameters need to be given/refined for successful localization and how much manual clean up is necessary? Is it applicable only to spheres or can it be extended to tubular structures or more complicated shapes? Additionally, is it possible to pick without a template by oversampling the membrane surface?
12. The deep learning-based picking has been illustrated on ribosomes only, as far as I can tell. Since particle picking is still one of the biggest challenges in CET (with the notable exception of ribosomes), it would be nice to see how the picking performs on other complexes in situ.
13. Are there plans to implement beam tilt and higher order aberration refinements? This could be helpful especially for beam image shift tomography.
14. The similarity score is only shown for the in vitro dataset. How does this behave for an in-situ dataset? Does the similarity score remain a useful metric when the SNR is even lower?
16. Why are only IMOD’s alignment routines included? Is it necessary to refine the IMOD alignment models manually like in etomo or are there additional layers of quality control? Are there options to use external programs like AreTomo?
17. The methods for strategies for 3D particle picking state the use of a contamination-detection step. The details of this step should be explained further.

Minor points:

- It might be helpful to introduce and contrast the terms of SP-CET, STA and CSPT that might be confusing for readers not familiar with the distinction.
- References 5-8 seem to not include an example of complexes within bacteria as stated.
- Supp. Fig. 1E should probably be captioned “tilt series alignment”?
- In the paragraph “Fully constrained refinement of particle poses”, the reduction of degrees of freedom to 6 parameters is described. To make it clearer, it might be helpful to reiterate the degrees of freedom for the original CSPT approach.
- For the full EMPIAR-10164 dataset, significantly fewer particles were used for the final reconstruction. Could you comment on why that might be? Less efficient particle picking or more stringent particle selection? Would the resolution be higher with the same number of particles?
- The text (page 11) mentions that Supp. Table 3 contains processing timing data, but it does not.
- Fig. 4B: Do you have the FSCs for the intermediate steps similar to Fig. 4A? It would be interesting to see the contributions of the different steps when going to the higher resolution regime.
- Which tilt series from the EMPIAR-10064 were used? VPP or defocus data? This should be clarified.
- The resolution of the consensus map from EMPIAR-10987 is 8.4 Å in the text, but 8.2 Å in the legend of Sup. Fig 5C.

Radostin Danev

Reviewer #2:

Remarks to the Author:

In this manuscript, Liu et al. ... Bartesaghi illustrate the development of a novel image processing platform (nextPYP) for averaging macromolecules and complexes from cryo-ET data. This is a timely and exciting development in the field because of the associated challenges with effective and efficient computational approaches for extracting complexes from tomography data for high-resolution structure determination. The authors provide multiple experimental examples from previously collected/published data to support the results produced by nextPYP, and to illustrate how it can be used broadly vs only for specialized target biological systems. This is an exciting addition to the field and could have a significant impact on the structural biology community! Only a few extremely minor points.

Page 14 and in figure legends: Italicize *M. pneumoniae*.

Page 19: Revise splitted into split (...in our region-based approach particles are splitted into groups based on their position).

Reviewer #3:

Remarks to the Author:

Liu et al. present a start-to-finish software package for single-particle cryo-electron tomography. This package incorporates existing algorithms from cisTEM and others which have been optimized for their proposed strategy of a modified form of constrained single particle tomography (CSPT). Tools included in the package include motion correction, CTF estimation, multiple particle pickers, 3D refinement, and 3D classification, among others. The authors point to several issues with current approaches including difficulty in moving between software packages, difficulty in particle picking within complex environments, and problems relating to the generation and storage of prohibitively large file sizes. The authors cite benefits of their program as being entirely self-contained, having minimal user input, fast processing speeds, and a reduced memory footprint.

Overall, this appears to be a nice addition to the existing software for determining structures from cryo-ET data, in particular in providing a one-stop-shop solution for users that want to process their data within one consistent pipeline. The manuscript is well written and presents the data convincingly, and could be considered appropriate paper for publishing in Nature Methods.

However, the manuscript could be more explicit in stating what the major novelties of the different new implementations, for example in classification and/or particle picking are. Also, the improved refinement strategies are nice, but seem to not exceed the resolutions that have been obtained by other software packages until now.

Major Points

- Many features of the software package are re-integrations of existing tools which have been optimized for this package. This includes motion correction, tilt series alignment, CTF estimation, size-based particle picker, geometry-based picking. The neural network picker has been optimized to remove labor-intensive labelling, now requiring only sparse annotation. While exciting and helpful, it is unclear exactly how innovative such an improvement really is. For 3D refinement and reconstruction, the package relies on algorithms within cisTEM. The authors implement an approach to relax global constraints on geometry - based on nearby particles, similar to M. Similarly, they optimize CTF refinement based on existing tools. 3D classification is based on cisTEM but with addition of constraints derived from their unique approach. Thus it should be made clear which algorithms are newly developed and which are implementations of previous algorithms. Overall, despite being implementations of existing tools, they have skilfully integrated these tools into a user-friendly package, making this processing strategy accessible to the cryo-EM community.
- Compared to their previous implementation of CSPT, which constrains exclusively orientations, this approach now also constrains translations. These translation constraints presumably depend on the accuracy of initial particle picks? It is unclear how this leads to an improvement in results. Many

available software packages also constrain translations relative to initial particle pick positions, so why is this was not the default behavior in previous CSPT?

- In Figure 3 (HIV-1 Gag), only the results for the first 9 frames are shown, but not beyond that. Does this imply that beyond the 9th frame, the data is not usable/was not used in their approach? Given that in vitro assemblies of retroviral particles can nowadays be solved entirely using SPA (for example see Highland et al., <https://doi.org/10.1073/pnas.2220545120>, and Stacey et al., <https://doi.org/10.1073/pnas.2220557120>), this would essentially further underscore that cryo-ET of such particles is not entirely necessary for obtaining high-resolution.
- One major selling point the authors mention for their software is the reduced storage footprint, by not working with full tomograms/subtomograms. This is indeed nice and important, but the comparisons to underscore this point are not always fully fair. For example, for EMPIAR-10164 the authors work with the super-resolution pixelsize, which is not necessary (and upon checking, has been also not been done in the original citations of this dataset). Given that the obtained resolution is so far away from super-resolution Nyquist, working with the physical pixelsize would be more appropriate. This significantly reduces the footprint.
- As the maps are directly calculated from the movie frames, what validation tools are in place to provide the user with info if the SNR per frame/movie has been too low for accurate movie refinement or self-tuned exposure weighting? This relates to the question with the use of 9 frames only in the HIV-1 Gag dataset.
- The constrained tilt classification is an interesting approach. However, it is not entirely clear how many projections are used in the classification, e.g. the entire extent of the tilt series or only a subset of tilts (as shown with the exposure weighting for the HIV-1 dataset). How does the user decide how many tilts to use? Also, does the range of tilts have an influence on the size of features that can be distinguished (i.e. at which point does the decreased SNR limit the classification).
- The particle picking implementations are shown on the three most used examples for subtomogram averaging, i.e. HIV-1 VLPs, apoferritin and ribosomes. Which kind of other geometries can be picked by the users, beyond spheres? For the semi-supervised deep-learning approach for particle identification, it would be relevant to provide some further examples to show how robust this works on something less easy than ribosomes.
- Software-related comment: NextPyp needs an additional admin user that is used for running the master/database/webserver daemon. As a consequence, one has to implement their own user management, and one does not make use of the Linux/Unix permission model. So this has security implications. This would not be an issue when one can assume that data access is only possible through

their software - which is not correct in our case, because we use the same storage servers, and users can access that data through the operating system. The data needs to be accessible by the users as well as the user running the service daemons NextPyP. In order to make it work, the permissions for data access on the operating system level need to be relaxed (everyone has rights access). This defeats the purpose of user permissions, or make it difficult to get the permissions "right".

In other words, there are two user/permission management systems, (1) from the OS, and (2) through the Web application, and these do not work well together. In our opinion, this is a main difficulty of NextPyP in an HPC environment, where users have access to the data through the operating system. NextPyP seems to be suitable when it is the only application running on some compute system, and all data access needs to go through their web interface. But this is hardly the case. Therefore, we found NextPyP difficult to install and use on a typical HPC cluster. Maybe the developers can provide some more info on how they see this apparent issue.

Minor points

- The primary citation for EMPIAR-10164 appears to be missing. For consistency, as EMPIAR-10499 is properly cited, the EMPIAR-10164 citation needs to be added as well.
- Page 11: "Averaging the scores from all the frames (after removing the offset for each tilt)....". Please specify what you mean with offset here.
- Fig 1B "patch-based", why are the patches not uniform? Rather should this not be slightly overlapping?
- Fig 1B. "size-based" unsure what this is depicting, perhaps a different image could be used to display this concept.
-
- Fig 3A. what is the color of the trajectories referring to?

Software-related:

- Typo: when setting up accounts, click "add group" a window pops up with a title "add user" (rather than add group).
- There is no mention in the manuscript that the program also handles direct single particle data, which could be of interest to many readers.
- In the GUI there is a small funnel shaped icon in the bottom right when searching for files to import. What does this button do? Hovering over it does not give any hint/tip.

Author Rebuttal to Initial comments

Reviewers' Comments:

Reviewer #1:

Remarks to the Author:

Liu and colleagues present a new end-to-end framework for processing, reconstruction, and 3D analysis of electron cryo-tomography data. The software has a user-friendly GUI-based interface and provides multiple interactive ways to visualize data, processing workflows, and results. Importantly, the package is all-inclusive and will give researchers a quick low-barrier entry option into the field of subtomogram analysis. Overall, this is an important contribution to the electron cryo-tomography data analysis ecosystem which at present lacks a unified processing solution and has relied on custom workflows using a mélange of software tools. The manuscript is well-written and gives mostly satisfactory descriptions of the algorithms and approaches used in the system. The work will be of particular interest to researchers who are already using cryo-tomography structural analysis or have been waiting for the technique to become more accessible before investing time into it. The prospects of improving the convenience and throughput of cryo-tomography will also intrigue the broader structural biology, disease studies, and drug development audiences. Therefore, I believe that the paper is a good match for Nature Methods and pending a few comments and questions, I enthusiastically recommend it for publication.

1. The version of nextPYP described in the paper uses mainly CPU computation. GPUs have proven to be significantly more efficient in terms of cost and overall speed for cryo-EM data analysis. With increasing dataset sizes and limited access to CPU-based clusters at many research institutions, GPUs may be the only processing option for many labs. Please add a comment in the paper about the possibilities or limitations of transitioning nextPYP to GPU compute.

The first release of nextPYP will only support GPUs for training the neural network models used for particle picking. All other operations will run on the CPU. This is because nextPYP uses cisTEM in the backend, which has been heavily optimized to run efficiently on CPUs. That being said, recent development versions of cisTEM are starting to expand support for GPUs and once this functionality is released, it will be possible for nextPYP to run on GPUs as well. We have commented on this topic in the Online Methods section.

2. Related to the above comment, please add a row in the Platform section of Supplementary Table 1 about GPU compute support.

Thanks for the suggestion. We added a new row to the Platform section in Supp. Table 1 indicating the type of GPU support provided by each package.

3. The region-based constrained geometry refinement relies on a good number and distribution of particles in the sample. This is not always the case in situ. How do you determine the number/size of regions?

Can they be automatically and/or non-uniformly distributed based on the local particle density?
Did you test different region sized on any of your datasets?
Is it possible to use more abundant particles like ribosomes for the region-based refinement and use these parameters for the particle alignment of other species similar to M?
Please comment on these points in the paper.

The number and size of patches is determined based on the amount of signal present in each region, which in turn is proportional to the molecular weight of the particles and their concentration. For bigger and more abundant particles, smaller regions can be chosen, while smaller or less concentrated particles need larger regions. Patches are currently selected uniformly in XYZ (i.e., 8x8x2 or 4x4x2), independent of the particle density. If a region doesn't have enough particles, refinement for that region will be skipped and the global alignment solution will be used. We did try increasing the number of regions beyond 8x8x4 in the datasets we analyzed, but saw only marginal improvements in resolution. In general, however, the use of finer grids may be beneficial for samples that undergo significant deformations during imaging. nextPYP does not currently support using more abundant particles to improve the alignment of other minority species the way M does. While so far we haven't come across any datasets that could take advantage of this feature, it will not be difficult to add this functionality in the future. We commented on all these points in the revision.

4. Does nextPYP support "hands-off" processing (on-the-fly or post-acquisition) by setting the necessary parameters initially and then letting it work through the data? If not, would it be possible in the future to develop nextPYP-based workflows that do not require interactive steps?

Yes, the Session's side of nextPYP is designed to do on-the-fly data processing during data collection (Results section, On-the-fly processing of tilt-series). Post-acquisition, nextPYP also supports the creation of "Workflows" consisting of user-defined sequences of processing blocks. When running in this mode, the user is asked to provide the required parameters at the beginning of the run (location of the data, pixel size, particle picking parameters, etc.), and all downstream blocks are executed automatically. We added a sentence to the Online Methods section describing this feature.

5. Why were the global reconstructions of the EMPIAR-10064 and EMPIAR-10499 not included in the results? In particular, a global reconstruction of EMPIAR-10499 comparing the performance of nextPYP and M would be very interesting.

We did not include the global reconstruction for EMPIAR-10064 because it was published previously in [Bouvette et al., Nat Comm, 2021, Supp. Figure 4]. The resolution of the global map for this dataset was 5.6 Å (<https://www.ebi.ac.uk/emdb/EMD-23357>) vs. M's 5.7 Å. For EMPIAR-10499, our global reconstruction has a resolution of 3.9 Å. This number is worse than M's 3.5 Å likely because: 1) our reconstruction was obtained using 25% fewer particles (18,135 vs. 24,202), and 2) we did not perform any particle cleaning or filtering since our main goal with this dataset was to benchmark our constrained classification routines. We included these details in the revision and presented the consensus map for 10499 in the new Supplementary Figure 6.

6. The reconstructions of three of the six datasets presented used a very narrow tilt range (-6 to +6 deg, Supplementary Table 3). Please add an explanation about that, the selection process of these tilt ranges, and the impact of the size of the processed tilt range on the processing time.

The selection of the number of tilts used for refinement was determined by measuring the relative contribution of each tilt to the final map. Once dose weighting is enabled, nextPYP produces plots similar to those shown in Fig. 3C where the contribution of each tilt is plotted as a function of the spatial frequency. Based on this, we only use tilts that contribute to high-resolution during refinement. The use of fewer tilts does speed up refinement, with computational complexity being linear in the number of projections. For example, using 1/10-th of the images (i.e., +/6° vs +/-60°), gives a 10x speed improvement during refinement. However, the overall speedup is typically lower, because all the tilts need to be extracted and used during 3D reconstruction. We added a sentence to the Results section explaining these details.

7. The particle cleaning uses a user-specified particle score cutoff. How is the optimal value selected? (visually?) Are there possible ways to determine the optimal cutoff value automatically?

For large particles like ribosomes, we often observe a bimodal distribution of scores and in these cases the selection of the cutoff can be automated as we showed in [Zhou et al., *Inv Prob*, 2019]. For more challenging targets, however, the similarity scores are not able to discriminate between properly aligned and poorly aligned particles and in these cases the cutoff must be specified manually. nextPYP routinely produces score histograms and displays them on the GUI to facilitate this process. We commented on this topic in the Online Methods section.

8. The constrained particle projection classification applies an empirical Gaussian upweighting of low tilt angle images which seems rather arbitrary. Can the image weighting be based on the frame scores from the 3D reconstruction (Fig. 3C)?

The rationale for using Gaussian weights was first introduced in [Bartesaghi et al., *Structure*, 2018] in the context of movie frame alignment in SPA. The advantage of this approach is that the weights can be precalculated and they do not need to be adjusted for each dataset. While it is possible to use data-specific weights derived from the scores, we saw that the effects of this on classification were negligible (likely because classification is mainly driven by lower resolution information). Moreover, when scores do not accurately reflect image alignment quality (e.g., in the case of small particles or low SNR samples), this approach can actually lower classification performance. For these reasons, we decided to go with the more conservative choice of using the Gaussian weights instead of using score-based weights. We added this discussion to the Online Methods section.

9. The “ab-initio” pose estimation approach is in fact reference-based. This is a bit confusing. What references were used and to what extent were they low-pass filtered?

Thanks for pointing this out, we changed "ab-initio" refinement to "reference-based" refinement. For 10164 and 10064 we used reconstructions and corresponding particle alignments obtained using traditional sub-volume averaging. For 10499, 10987, and 11273 we used our reference-based refinement approach using references downloaded from the EMDB and filtered to 16Å, 16Å, and 6Å, respectively. We added a new row to Supp. Table 3 with this information.

10. It is stated that the size-based particle picking approach can facilitate particle picking of large complexes like ribosomes in-situ, yet the ribosome datasets shown here were picked with the deep learning approach. It would be interesting to see a comparison between the two approaches for in vitro and in situ data.

We added the new Supp. Figure 9 and Supp. Table 5 showing a comparison between the size-based and neural network approaches when applied to in vitro (EMPIAR-10304) and in situ (EMPIAR-10499) datasets. We used manually selected particles from both datasets as ground-truth and calculated Precision, Recall and F1 metrics in each case. The performance of the neural network based approach is somewhat better in both cases, but the size-based approach requires less user input and runs faster because it doesn't require labels or time for training. In general, the decision of which approach is most appropriate will depend on the accuracy requirements and computational considerations of each project. The philosophy behind nextPYP is to provide different choices for particle picking and let users decide which approach works best for their data. We added a paragraph to the Online Methods section describing the new experiments comparing the two particle picking approaches.

11. For the virion segmentation. What parameters need to be given/refined for successful localization and how much manual clean up is necessary? Is it applicable only to spheres or can it be extended to tubular structures or more complicated shapes? Additionally, is it possible to pick without a template by oversampling the membrane surface?

The only parameter needed for virion segmentation is the approximate radius of the viruses. The amount of manual clean up depends on the contrast and crowdedness of the tomograms. For EMPIAR-10164, for example, no manual cleanup is required. The geometry-based strategy for virion segmentation has two parts: 1) detection of the virion centers, and 2) segmentation of the corresponding membranes. The center detection part in nextPYP can currently only handle round objects, but since the algorithm is based on the Hough transform, it could potentially be adapted to search for ellipsoidal objects as well. The algorithm we use for segmentation is based on energy minimization and it can also be generalized to arbitrary shapes, but when used within nextPYP, it can only segment membranes that are contained within two concentric spheres (i.e., within a min and max radius from the origin). Supporting other shapes will require making significant changes to the UI, which we may consider doing in future versions. Once the segmentation is obtained, nextPYP can produce equally spaced positions on the surface of the virions without the need of a template (this mode is called "uniform" picking). We added these details to the Results section.

12. The deep learning-based picking has been illustrated on ribosomes only, as far as I can tell. Since particle picking is still one of the biggest challenges in CET (with the notable exception of ribosomes), it would be nice to see how the picking performs on other complexes in situ.

This is correct, our particle picking algorithm based on deep learning has only been tested on ribosomes from in-vitro and in-situ datasets (experiments shown in [Huang et al., ECCV 2022] and the new Supp. Figure 9 and Supp. Table 5). To address this point, we now measured the performance of our particle picking strategy on a more challenging in situ dataset of *Chlamydomonas reinhardtii* cells (EMPIAR-10694) containing RuBisCO particles (molecular weight ~560kDa). We picked 86 particles by hand and used them to train our deep neural network approach. We used the trained model to pick a total of 35,352 particles which were subjected to 3D refinement and produced a 12 Å resolution map (vs. EMD-3694's 16.5Å), demonstrating the ability of this approach to pick non-ribosome complexes imaged in situ. The new data is presented in Supp. Figure 8 and is described in Online Methods (sub-section "Semi-supervised picking based on deep neural networks").

13. Are there plans to implement beam tilt and higher order aberration refinements? This could be helpful especially for beam image shift tomography.

nextPYP currently uses a version of cisTEM (v1.0) that does not support correction of beam tilt or higher order aberrations. Newer development versions of cisTEM, however, do incorporate routines for beam tilt correction and while we plan to upgrade the code to use the newer version, this will not be part of the initial release of nextPYP.

14. The similarity score is only shown for the in vitro dataset. How does this behave for an in-situ dataset? Does the similarity score remain a useful metric when the SNR is even lower?

Indeed, the reliability of the similarity scores is dependent on the SNR of the projections (lower SNR = less reliability). Since the in-situ ribosome datasets EMPIAR-10987 and EMPIAR-10499 were used to validate our 3D classification strategy, we did not refine the individual movie frames for these datasets, so we don't have in situ examples of per-frame similarity scores. However, we did calculate scores on a *per-tilt* basis for these datasets and these do provide a useful metric of data quality. We included an example of per-tilt similarity scores obtained for EMPIAR-10987 in the new Supplementary Figure 7, showing that the scores correctly capture the relative contribution of each tilt as a function of the tilt-angle and the accumulated exposure. For smaller complexes or individual frames with lower SNR, however, the scores may not accurately reflect the quality of the data. We commented on this topic in the revision.

16. Why are only IMOD's alignment routines included? Is it necessary to refine the IMOD alignment models manually like in etomo or are there additional layers of quality control? Are there options to use external programs like AreTomo?

We only included IMOD routines for tilt-series alignment and reconstruction because we are yet to find cases where these routines fail. nextPYP adds additional layers on top of IMOD to make

these routines more robust and less prone to error. The current version of nextPYP does not provide the option to incorporate external programs for tilt-series alignment, like AreTomo, partly because IMOD is already very reliable and partly because AreTomo's reliance on GPUs may limit throughput when trying to align thousands of tilt-series in parallel (which can be done easily using CPU resources). That being said, we do plan to support third party tools for tilt-series alignment and other operations in the future.

17. The methods for strategies for 3D particle picking state the use of a contamination-detection step. The details of this step should be explained further.

We expanded the description of the contamination-detection step in Online Methods.

Minor points:

- It might be helpful to introduce and contrast the terms of SP-CET, STA and CSPT that might be confusing for readers not familiar with the distinction.

Thanks for the suggestion, we clarified these terms in the revision and now refer to CSPT as "constrained SP-CET" to reduce the number of acronyms and avoid confusion.

- References 5-8 seem to not include an example of complexes within bacteria as stated.

These references were only intended to cite work done on eukaryotic cells (not on bacterial samples). We have now added separate references for the bacterial samples.

- Supp. Fig. 1E should probably be captioned "tilt series alignment"?

Yes, thanks for noticing this, we made the change in the revision.

- In the paragraph "Fully constrained refinement of particle poses", the reduction of degrees of freedom to 6 parameters is described. To make it clearer, it might be helpful to reiterate the degrees of freedom for the original CSPT approach.

In the original CSPT approach, the degrees of freedom were 3 angles *per particle* + two translations *per particle projection*. If using tilt-series with 41 tilts, for example, this represents a reduction in the number of free parameters per particle from $3+2*41=85$ down to only 6. We clarified this in the revision.

- For the full EMPIAR-10164 dataset, significantly fewer particles were used for the final reconstruction. Could you comment on why that might be? Less efficient particle picking or more stringent particle selection? Would the resolution be higher with the same number of particles?

The main reason for the lower number of particles is that we did more stringent particle selection. The starting number of particles was 425,424 (Supp. Table 3) and after filtering we only kept 109,496 particles. Using more particles would not have increased resolution because

this dataset was acquired with an aperture that physically low pass filtered the data to 3 Å (which is already the nominal resolution of our reconstruction).

- The text (page 11) mentions that Supp. Table 3 contains processing timing data, but it does not.

Thanks for noticing this, we removed that reference to Supp. Table 3 from the text.

- Fig. 4B: Do you have the FSCs for the intermediate steps similar to Fig. 4A? It would be interesting to see the contributions of the different steps when going to the higher resolution regime.

Yes, the baseline reconstruction of apoferritin after fully constrained refinement had a resolution of 4.4 Å. Improvements in resolution corresponding to the different steps were as follows: data-driven exposure weighting (3.9 Å), region-based constrained refinement (2.1 Å), particle-based CTF refinement (2.0 Å), and movie frame refinement (1.8 Å). We added a new panel to Fig. 4B showing the new FSC curves with the intermediate refinement steps.

- Which tilt series from the EMPIAR-10064 were used? VPP or defocus data? This should be clarified.

We used the 4 tilt-series from the defocus data (mixedCTEM_tomo). We clarified this in the text and Supp. Table 2.

- The resolution of the consensus map from EMPIAR-10987 is 8.4 Å in the text, but 8.2 Å in the legend of Sup. Fig 5C.

Thanks for noticing this, the resolution should have been reported as 8.4 Å. We fixed this in the legend and also added more tick marks to the x-axis of the plot shown in Supp. Fig 5C.

Radostin Danev

Reviewer #2:

Remarks to the Author:

In this manuscript, Liu et al. ... Bartesaghi illustrate the development of a novel image processing platform (nextPYP) for averaging macromolecules and complexes from cryo-ET data. This is a timely and exciting development in the field because of the associated challenges with effective and efficient computational approaches for extracting complexes from tomography data for high-resolution structure determination. The authors provide multiple experimental examples from previously collected/published data to support the results produced by nextPYP, and to illustrate how it can be used broadly vs only for specialized target biological systems.

This is an exciting addition to the field and could have a significant impact on the structural biology community! Only a few extremely minor points.

Page 14 and in figure legends: Italicize *M. pneumoniae*.

Page 19: Revise splitted into split (...in our region-based approach particles are splitted into groups based on their position).

Thank you for the positive feedback and for noticing these issues, we fixed them in the revision.

Reviewer #3:

Remarks to the Author:

Liu et al. present a start-to-finish software package for single-particle cryo-electron tomography. This package incorporates existing algorithms from cisTEM and others which have been optimized for their proposed strategy of a modified form of constrained single particle tomography (CSPT). Tools included in the package include motion correction, CTF estimation, multiple particle pickers, 3D refinement, and 3D classification, among others. The authors point to several issues with current approaches including difficulty in moving between software packages, difficulty in particle picking within complex environments, and problems relating to the generation and storage of prohibitively large file sizes. The authors cite benefits of their program as being entirely self-contained, having minimal user input, fast processing speeds, and a reduced memory footprint.

Overall, this appears to be a nice addition to the existing software for determining structures from cryo-ET data, in particular in providing a one-stop-shop solution for users that want to process their data within one consistent pipeline. The manuscript is well written and presents the data convincingly, and could be considered appropriate paper for publishing in Nature Methods.

However, the manuscript could be more explicit in stating what the major novelties of the different new implementations, for example in classification and/or particle picking are. Also, the improved refinement strategies are nice, but seem to not exceed the resolutions that have been obtained by other software packages until now.

Major Points

- Many features of the software package are re-integrations of existing tools which have been optimized for this package. This includes motion correction, tilt series alignment, CTF estimation, size-based particle picker, geometry-based picking. The neural network picker has been optimized to remove labor-intensive labelling, now requiring only sparse annotation. While exciting and helpful, it is unclear exactly how innovative such an improvement really is. For 3D refinement and reconstruction, the package relies on algorithms within cisTEM. The authors implement an approach to relax global constraints on geometry - based on nearby particles, similar to M. Similarly, they optimize CTF refinement based on existing tools. 3D classification is

based on cisTEM but with addition of constraints derived from their unique approach. Thus it should be made clear which algorithms are newly developed and which are implementations of previous algorithms. Overall, despite being implementations of existing tools, they have skilfully integrated these tools into a user-friendly package, making this processing strategy accessible to the cryo-EM community.

We realize that the distinction between re-implementations and novel methods was not clear in the original submission. The following strategies are re-implementations or adaptations of existing methods:

- Motion correction (based on unblur, optimized to work on tilt-series)
- Tilt-series alignment (based on IMOD, with added robustness)
- CTF estimation (based on our previous approach [Bouvette et al., Nat Comm, 2021])
- Geometry-based picking (re-implementation of methods originally presented in [Bartesaghi et al., 2008])
- Neural network picker (re-implementation based on [Huang et al., ECCV, 2022] that incorporates an easy-to-use GUI for training, inference and inspection of results).

The following are novel methods and features introduced in this manuscript:

- Size-based 3D particle picking with contamination detection
- Stack-less, scalable framework for constrained SP-CET refinement
- Reference-based alignment from projections (without the need to generate sub-volumes)
- Fully-constrained, region-based and particle-based CTF refinement (similar strategies were implemented in M using Relion primitives)
- Movie-frame refinement, data-driven exposure weighting and per-frame reconstruction
- Constrained approach for 3D classification from 2D projections

We added a new Supplementary Table 4 with a summary of re-implementations vs. new methods introduced in this manuscript.

• Compared to their previous implementation of CSPT, which constrains exclusively orientations, this approach now also constrains translations. These translation constraints presumably depend on the accuracy of initial particle picks? It is unclear how this leads to an improvement in results. Many available software packages also constrain translations relative to initial particle pick positions, so why is this was not the default behavior in previous CSPT?

The translation constraints themselves do not depend on the accuracy of the initial particle picks. Perhaps we should clarify that there are two separate translations involved here: 1) translations in 3D applied to each particle, and 2) translations in 2D applied to individual particle projections. While the reviewer seems to be referring to 3D translations, our fully constrained approach relates to changes in 2D translations between tilts when particles are rotated and translated in 3D. The reason this was not the default behavior in the original CSPT algorithm is that this approach required more careful bookkeeping and additional development, which we have done for this manuscript. Similar to other packages, we also use a tolerance parameter that limits how far away from the picked positions the search for 3D translations should extend. This parameter, however, is independent of the constraints. Our comment about "improved resolution" is associated with the reduction in the number of free parameters, which helps lower

the dimensionality of the search space making refinement less prone to overfitting (see response to Reviewer 1). We clarified these points in the Results section.

- In Figure 3 (HIV-1 Gag), only the results for the first 9 frames are shown, but not beyond that. Does this imply that beyond the 9th frame, the data is not usable/was not used in their approach? Given that in vitro assemblies of retroviral particles can nowadays be solved entirely using SPA (for example see Highland et al., <https://doi.org/10.1073/pnas.2220545120>, and Stacey et al., <https://doi.org/10.1073/pnas.2220557120>), this would essentially further underscore that cryo-ET of such particles is not entirely necessary for obtaining high-resolution.

The plot in Figure 3 includes information for *all* the frames from *all* the tilts (not only the first 9). The reason frames for tilts beyond 9 degrees are not visible is because their contribution is very small and only the white outlines between the bands are seen in the plot. We clarified this in the legend of Figure 3. Regarding the point about the usefulness of images beyond this range, we note that while higher tilt-images may not contribute high-resolution information to the final map, 1) they are helpful in producing 3D tomograms that are used for particle picking, and 2) the wider tilt-range facilitates reference-based particle alignment and classification. Without the information from higher tilt angles, it would be challenging to pick particles and find their initial orientations as needed for the downstream processing. That being said, we agree with the reviewer that for in vitro assemblies of retroviral particles like EMPIAR-10164, cryo-ET may not strictly be needed (as shown in the references cited by the reviewer). For more challenging in situ samples, however, it will be difficult to produce structures exclusively using SPA for the reasons stated above. We commented on the contribution of low tilts to the final map in the Results section.

- One major selling point the authors mention for their software is the reduced storage footprint, by not working with full tomograms/subtomograms. This is indeed nice and important, but the comparisons to underscore this point are not always fully fair. For example, for EMPIAR-10164 the authors work with the super-resolution pixel size, which is not necessary (and upon checking, has been also not been done in the original citations of this dataset). Given that the obtained resolution is so far away from super-resolution Nyquist, working with the physical pixel size would be more appropriate. This significantly reduces the footprint.

We agree that using the super-resolution pixel size is not necessary for this dataset. While we did mention in the Methods section that the super-resolution data was used, this only applied to the pre-processing steps. All subsequent 3D refinement steps were done with binned 2x data (1.35 Å/pixel) using a box size of 384 pixels. For example, the 25 TB number reported for the full dataset (43 tilt-series) in the Results section, was calculated by adding the space required to extract the sub-volumes (22.5 TB) and the corresponding particle stacks (2.5 TB) at 2x binning (1.35 Å/pixel) using floating point precision (4-bytes per pixel/voxel):

- 109,496 sub-volumes at 384³ box size: $109,496 \times 384^3 \times 4 / 1024^4 = 22.5$ TB

- 109,496 sub-volumes represented with 41 tilts each: $109,496 \times 41 \times 384^2 \times 4 / 1024^4 = 2.5$ TB

Moreover, we note that these calculations only include particles that contributed to the final map (not the starting 425,424 particles), and they do not include storage space needed to do the

movie frame analysis which will result in an additional 10x increase in storage. The numbers reported in Supp. Table 2 were also calculated using the binned 2x data (1.35 Å/pixel), as follows:

- $14,480 \text{ particles} \times 384^3 \times 4 / 1024^4 = 3 \text{ TB}$

- $4,287,532 \text{ projections} \times 384^2 \times 4 / 1024^4 = 2.3 \text{ TB}$

We clarified the choice of pixel sizes in the text to prevent confusion.

- As the maps are directly calculated from the movie frames, what validation tools are in place to provide the user with info if the SNR per frame/movie has been too low for accurate movie refinement or self-tuned exposure weighting? This relates to the question with the use of 9 frames only in the HIV-1 Gag dataset.

The way we assess the presence of overfitting is by looking at the shape of the FSC curves as presented in [Penczek, Methods Enzymol., 2010]. Reference based refinement in cisTEM is done by calculating similarity scores between the raw 2D images and projections of the 3D reference, only including frequency components up to a user-specified maximum resolution. If the resulting FSC curve after refinement has a steep falloff at the maximum frequency used for refinement, this will indicate the presence of overfitting and map quality will worsen. To prevent overfitting during movie-frame refinement, we impose stronger spatial constraints on the movement of neighboring particles and enforce temporal regularity of particle trajectories. Regarding data-driven exposure weighting, the way we assess whether the score numbers are accurate is by looking at the shape of the per-frame score curves. In the overfitting regime, the curves will appear noisy and will not show the characteristic bell shape seen in Figure 3B-D. When this occurs, application of exposure weighting will actually lower the quality of the map. To prevent this behavior, consecutive frames can be averaged prior to frame alignment or scores can be averaged across multiple tilt-series to increase the SNR. We commented on these points in the Online Methods section: "Movie-frame refinement and self-tuning exposure weighting".

- The constrained tilt classification is an interesting approach. However, it is not entirely clear how many projections are used in the classification, e.g. the entire extent of the tilt series or only a subset of tilts (as shown with the exposure weighting for the HIV-1 dataset). How does the user decide how many tilts to use? Also, does the range of tilts have an influence on the size of features that can be distinguished (i.e. at which point does the decreased SNR limit the classification).

Since classification is mainly driven by the lower resolution components of the data, we use *all* the tilts in our constrained classification approach. While nextPYP does provide the ability to use a smaller subset of tilts for classification, we haven't done systematic tests on how the range of tilts affects classification performance. It is true, however, that a very narrow range of projections may not provide sufficient SNR to allow accurate classification. We clarified this point in the text.

- The particle picking implementations are shown on the three most used examples for subtomogram averaging, i.e. HIV-1 VLPs, apoferritin and ribosomes. Which kind of other geometries can be picked by the users, beyond spheres? For the semi-supervised

deep-learning approach for particle identification, it would be relevant to provide some further examples to show how robust this works on something less easy than ribosomes.

The geometry-based picking approach in the current version of nextPYP only supports the detection of round objects (see response to Reviewer 1). While no tools are currently available in nextPYP to pick particles from more complex geometries, the GUI does provide the ability to pick particles interactively using the web-interface, and these positions can be used to train a neural network model that can later be applied to multiple tomograms. For the semi-supervised deep-learning approach, we now provide additional examples for in vitro and in situ datasets of ribosomes (new Supplementary Figure 9 and Supplementary Table 5), and a more challenging cellular dataset containing RuBisCO particles (new Supplementary Figure 8, see also response to Reviewer 1).

- Software-related comment: NextPyp needs an additional admin user that is used for running the master/database/webserver daemon. As a consequence, one has to implement their own user management, and one does not make use of the Linux/Unix permission model. So this has security implications. This would not be an issue when one can assume that data access is only possible through their software - which is not correct in our case, because we use the same storage servers, and users can access that data through the operating system. The data needs to be accessible by the users as well as the user running the service daemons NextPyp. In order to make it work, the permissions for data access on the operating system level need to be relaxed (everyone has rights access). This defeats the purpose of user permissions, or make it difficult to get the permissions "right".

In other words, there are two user/permission management systems, (1) from the OS, and (2) through the Web application, and these do not work well together. In our opinion, this is a main difficulty of NextPyp in an HPC environment, where users have access to the data through the operating system. NextPyp seems to be suitable when it is the only application running on some compute system, and all data access needs to go through their web interface. But this is hardly the case. Therefore, we found NextPyp difficult to install and use on a typical HPC cluster. Maybe the developers can provide some more info on how they see this apparent issue.

Indeed, all jobs in nextPYP run under a unique "service" account. Since service accounts can be assigned lower permission levels than regular user accounts, this can actually result in improved security and is what motivated our decision to use service accounts in nextPYP. However, as the reviewer correctly points out, this means that any files generated by the service account will need to be readable by other Linux users, for example, to allow them to access the results from outside nextPYP or export data to other packages, etc.

There are two solutions to this problem:

1. Linux permissions can be configured so that only the service account and individual users have access to the files produced by nextPYP. This requires creating new Linux groups with each user and the service account being the only members. Since nextPYP uses a separate folder to store the files for each user, group ownership for this folder can be changed to the new groups (chown :new_group). By assigning this folder the "setgid"

bit (chmod g+s), every new file or folder generated by nextPYP under the user folder will be accessible only by the user and the service account. In cases where multiple users need to have access to the data, additional users can be added to the newly created groups. This ensures that any files written by nextPYP will only be accessible by members of the group. In some systems, it may be possible to use the standard "private user group" (that has the same name as the username), instead of creating new groups. In this case, the service account will need to be added to the private group for each user and ownership of the user folder be changed to the private group.

2. The second solution is to run a separate instance of nextPYP for each Linux user. This will ensure a 1-to-1 mapping between users of the application and Linux accounts and will not require the creation or modification of any Linux groups. In setups that have many users, however, this configuration will require a higher level of maintenance from system administrators (compared to having a single instance of nextPYP).

Both solutions will address the concerns raised by the reviewer regarding restricting access to the data. We added a comment in the Online Methods and a new section in the software documentation describing these solutions.

Minor points

- The primary citation for EMPIAR-10164 appears to be missing. For consistency, as EMPIAR-10499 is properly cited, the EMPIAR-10164 citation needs to be added as well.

We added [Schur et al, Science, 2015] to the references.

- Page 11: "Averaging the scores from all the frames (after removing the offset for each tilt)....". Please specify what you mean with offset here.

This "offset" refers to the average score for all the frames in a given tilt image. As can be seen in Figure 3B, there is an overall reduction in the score values with each subsequent tilt due to the effects of radiation damage. By removing this "offset" and averaging over all the tilts, we can get a clearer picture of what the scores per frame look like (Fig. 3D). We clarified this in the revision.

- Fig 1B "patch-based", why are the patches not uniform? Rather should this not be slightly overlapping?

The patches in our implementation are all the same size and non-overlapping. Fig. 1B shows the position of the patches after refinement, which may give the impression that the patches are not uniform. We clarified this in the text and the legend of Fig. 1.

- Fig 1B. "size-based" unsure what this is depicting, perhaps a different image could be used to display this concept.

This image shows a tomographic slice with yellow points representing particles picked using our size-based approach. To better show this, we now show a zoomed in version of this image in Fig 1B.

- Fig 3A. what is the color of the trajectories referring to?

The color indicates the frame number from 1 (red) to 8 (yellow). We added a colorbar to Fig. 3A to illustrate this.

Software-related:

- Typo: when setting up accounts, click "add group" a window pops up with a title "add user" (rather than add group).

Thanks for noticing this, we fixed this typo in the gitlab repository.

- There is no mention in the manuscript that the program also handles direct single particle data, which could be of interest to many readers.

Indeed, nextPYP can also handle single-particle data, but since this was not the focus of the manuscript, we originally decided to leave it out. We added a new section to Online Methods: "Support for single-particle data processing", describing single-particle functionality in nextPYP.

- In the GUI there is a small funnel shaped icon in the bottom right when searching for files to import. What does this button do? Hovering over it does not give any hint/tip.

The purpose of this button is to filter file names displayed in the file browser according to a wildcard specified by the user. For example, if the user wants to count how many .mrc files are in a given folder, they can provide the "*.mrc" extension in the text dialog and press the "filter/funnel" button to get the total count of .mrc files in the folder.

Decision Letter, first revision:

Dear Alberto,

Thank you for submitting your revised manuscript "nextPYP: a comprehensive and scalable platform for characterizing protein variability in-situ using single-particle cryo-electron tomography" (NMETH-A52074A). It has now been seen by the original referees and their comments are below. The reviewers find that the paper has improved in revision, and therefore we'll be happy in principle to publish it in Nature Methods, pending minor revisions to comply with our editorial and formatting guidelines.

TRANSPARENT PEER REVIEW

Nature Methods offers a transparent peer review option for new original research manuscripts submitted from 17th February 2021. We encourage increased transparency in peer review by publishing the reviewer comments, author rebuttal letters and editorial decision letters if the authors agree. Such peer review material is made available as a supplementary peer review file. Please state in the cover letter 'I wish to participate in transparent peer review' if you want to opt in, or 'I do not wish to participate in transparent peer review' if you don't. Failure to state your preference will result in delays in accepting your manuscript for publication.

ORCID

Sincerely,
Rita

Rita Strack, Ph.D.
Senior Editor
Nature Methods

Reviewer #1 (Remarks to the Author):

The authors have addressed all raised issues and answered all questions satisfactorily. In the process, the manuscript was improved substantially and I recommend it for publication.

Radostin Danev

Reviewer #3 (Remarks to the Author):

The authors have carefully addressed all comments raised in the previous revision round. I have no further questions or comments and recommend the manuscript for publication in Nature Methods.

Author Rebuttal, first revision:

We thank both reviewers for their time and positive feedback. On behalf of all the authors,
Alberto Bartesaghi, PhD

Final Decision Letter:

Dear Alberto,

I am pleased to inform you that your Article, "nextPYP: a comprehensive and scalable platform for characterizing protein variability in-situ using single-particle cryo-electron tomography", has now been accepted for publication in Nature Methods. Your paper is tentatively scheduled for publication in our November print issue, and will be published online prior to that. The received and accepted dates will be March 22, 2023 and Sept 12, 2023. This note is intended to let you know what to expect from us over the next month or so, and to let you know where to address any further questions.

Over the next few weeks, your paper will be copyedited to ensure that it conforms to Nature Methods style. Once your paper is typeset, you will receive an email with a link to choose the appropriate publishing options for your paper and our Author Services team will be in touch regarding any additional information that may be required.

You will receive a link to your electronic proof via email with a request to make any corrections within 48 hours. If, when you receive your proof, you cannot meet this deadline, please inform us at rjsproduction@springernature.com immediately.

Please note that *Nature Methods* is a Transformative Journal (TJ). Authors may publish their research with us through the traditional subscription access route or make their paper immediately open access through payment of an article-processing charge (APC). Authors will not be required to make a final decision about access to their article until it has been accepted. [Find out more about Transformative Journals](https://www.springernature.com/gp/open-research/transformative-journals)

Your paper will now be copyedited to ensure that it conforms to Nature Methods style. Once proofs are generated, they will be sent to you electronically and you will be asked to send a corrected version within 24 hours. It is extremely important that you let us know now whether you will be difficult to

contact over the next month. If this is the case, we ask that you send us the contact information (email, phone and fax) of someone who will be able to check the proofs and deal with any last-minute problems.

If, when you receive your proof, you cannot meet the deadline, please inform us at rjsproduction@springernature.com immediately.

Once your manuscript is typeset and you have completed the appropriate grant of rights, you will receive a link to your electronic proof via email with a request to make any corrections within 48 hours. If, when you receive your proof, you cannot meet this deadline, please inform us at rjsproduction@springernature.com immediately.

Once your paper has been scheduled for online publication, the Nature press office will be in touch to confirm the details.

Once your paper has been scheduled for online publication, the Nature press office will be in touch to confirm the details.

Content is published online weekly on Mondays and Thursdays, and the embargo is set at 16:00 London time (GMT)/11:00 am US Eastern time (EST) on the day of publication. If you need to know the exact publication date or when the news embargo will be lifted, please contact our press office after you have submitted your proof corrections. Now is the time to inform your Public Relations or Press Office about your paper, as they might be interested in promoting its publication. This will allow them time to prepare an accurate and satisfactory press release. Include your manuscript tracking number NMETH-A52074B and the name of the journal, which they will need when they contact our office.

About one week before your paper is published online, we shall be distributing a press release to news organizations worldwide, which may include details of your work. We are happy for your institution or funding agency to prepare its own press release, but it must mention the embargo date and Nature Methods. Our Press Office will contact you closer to the time of publication, but if you or your Press Office have any inquiries in the meantime, please contact press@nature.com.

To assist our authors in disseminating their research to the broader community, our SharedIt initiative provides you with a unique shareable link that will allow anyone (with or without a subscription) to read

the published article. Recipients of the link with a subscription will also be able to download and print the PDF.

Nature Portfolio journals [encourage authors to share their step-by-step experimental protocols](https://www.nature.com/nature-research/editorial-policies/reporting-standards#protocols) on a protocol sharing platform of their choice. Nature Portfolio 's Protocol Exchange is a free-to-use and open resource for protocols; protocols deposited in Protocol Exchange are citable and can be linked from the published article. More details can found at www.nature.com/protocolexchange/about.

Best regards,
Rita

Rita Strack, Ph.D.
Senior Editor
Nature Methods